# An instrument for quantifying heterogeneous ice nucleation in multiwell plates using infrared emissions to detect freezing

**Alexander D. Harrison[1], Thomas F. Whale[1\*], Rupert Rutledge[2], Stephen Lamb[2], Mark D. Tarn[1], Grace C. E. Porter[1], Michael Adams[1], James B. McQuaid[1], George J. Morris[2] and Benjamin J. Murray[1]**

[1]Institute for Climate and Atmospheric Science, School of Earth and Environment, University of Leeds, Woodhouse

Lane, Leeds, LS2 9JT, UK

[2]Asymptote Ltd., GE Healthcare, Sovereign House, Cambridge, CB24 9BZ, UK

[\*] Now at School of Chemistry, University of Leeds, Woodhouse Lane, Leeds, LS2 9JT, UK.

*Correspondence to*: A. D. Harrison (ee11ah@leeds.ac.uk) and B. J. Murray (b.j.murray@leeds.ac.uk)

**Abstract**
Low concentrations of ice nucleating particles (INPs) are thought to be important for the properties of mixed-
phase clouds, but their detection is challenging. Hence, there is a need for instruments where INP concentrations
of less than 0.01 L$^{-1}$ can be routinely and efficiently determined. The use of larger volumes of suspension in drop
assays increases the sensitivity of an experiment to rarer INPs or rarer active sites due to the increase in aerosol
or surface area of particulates per droplet. Here we describe and characterise the InfraRed-Nucleation by
Immersed Particles Instrument (IR-NIPI), a new immersion freezing assay that makes use of IR emissions to
determine the freezing temperature of individual 50 μL droplets each contained in a well of a 96-well plate. Using
an IR camera allows the temperature of individual aliquots to be monitored. Freezing temperatures are determined
by detecting the sharp rise in well temperature associated with the release of heat caused by freezing. In this paper
we first present the calibration of the IR temperature measurement, which makes use of the fact that following ice
nucleation aliquots of water warm to the ice-liquid equilibrium temperature (i.e. 0°C when water activity is ~1),
which provides a point of calibration for each individual well in each experiment. We then tested the temperature
calibration using ~100 μm chips of K-feldspar, by immersing these chips in 1 μL droplets on an established cold
stage (μL-NIPI) as well as in 50 μL droplets on IR-NIPI; the results were consistent with one another indicating
no bias in the reported freezing temperature. In addition we present measurements of the efficiency of the mineral
dust NX-illite and a sample of atmospheric aerosol collected on a filter in the city of Leeds. NX-illite results are
consistent with literature data and the atmospheric INP concentrations were in good agreement with the results
from the μL-NIPI instrument.  This demonstrates the utility of this approach, which offers a relatively high
throughput of sample analysis and access to low INP concentrations.
**1    Introduction**
Cloud droplets can freeze homogeneously below about -33°C (Herbert et al., 2015), but the presence of ice-
nucleating particles (INPs) can induce freezing at much warmer temperatures (Kanji et al., 2017). The glaciation
of clouds at these warmer temperatures has a substantial impact on a cloud's reflective properties, lifetime and
therefore the overall climate of the planet, but is poorly represented in many models (Hoose and Möhler, 2012;
Vergara-Temprado et al., 2018). INPs can cause nucleation through a number of pathways (Vali et al., 2015), but
in mixed-phase clouds it is thought that the pathways where particles become immersed in droplets is most
important (Hande and Hoose, 2017; Hoose et al., 2010; Murray et al., 2012). Even small concentrations of INPs
can influence cloud properties; for example, in a modelling study of Southern Ocean shallow mixed-phase clouds,
Vergara-Temprado et al. (2018) showed that while concentrations of INPs greater than ~1 $L^{-1}$ cause profound
changes in cloud properties, clouds are sensitive to concentrations many orders of magnitude smaller.
The ability to quantify INP spectra (INP concentrations as a function of temperature) and test the efficiency of
proxy materials for ice-nucleating efficiency is invaluable for improving our understanding of cloud glaciation
and developing computationally inexpensive parameterisations for atmospheric models. However it is not a trivial
task, in part because INP concentrations are low ($<0.1 L^{-1}$) (DeMott et al., 2010) and the sites on the surfaces which
cause nucleation at warm temperatures (Vali, 2014; Whale et al., 2017) are rare. There are several different
methods of conducting ice nucleation experiments that include Continuous Flow Diffusion Chambers (CFDC's)
(e.g. Garimella et al., 2016; Kanji and Abbatt, 2009; Kohn et al., 2016; Rogers et al., 2001; Salam et al., 2006;
Stetzer et al., 2008), cloud expansion chambers (e.g. Cotton et al., 2007; Niemand et al., 2012), wind tunnels (e.g.
Diehl and Mitra, 1998; Pitter and Pruppacher, 1973) and droplet freezing assays (e.g. Beall et al., 2017; Budke
and Koop, 2015; Häusler et al., 2018; Knopf and Alpert, 2013; Murray et al., 2011; Vali, 2008; Whale et al.,
2015). Each of these methods has its limitations and advantages which must be understood and accounted for
when conducting an experiment and interpreting the results. For example CFDCs cannot be used for
measurements at temperatures warmer than about -11°C but they do allow for specific saturation conditions to be
controlled, something which other instruments cannot achieve. For more information on the capabilities and
limitations of the various techniques see the comprehensive reviews and intercomparisons conducted by Hiranuma
*et al*. (2015) and (DeMott et al., 2018). Haüsler *et al*. (2018) also presents a summary of the features of various
techniques.
A significant challenge in sampling INPs in the atmosphere is their low concentration. At present there is a dearth
of published, atmospherically relevant, INP measurements globally (Kanji et al., 2017; Vergara-Temprado et al.,
2017). Not only is the global spatial and temporal coverage of INPs inadequate, but the range of activation
temperatures and INP concentrations covered in any one set of measurements is typically limited.  No single
instrument has the capability of measuring INP concentrations over the full range of conditions relevant to mixed-
phase clouds.  Online instruments, such as CFDCs, can measure over a wide range of relevant conditions but their
detection limit is limited to $\sim 10^{-1}$ L$^{-1}$ (Al-Naimi and Saunders, 1985; DeMott et al., 2010; Eidhammer et al., 2010;
Prenni et al., 2009). This can be improved with aerosol concentrators (Prenni et al., 2013; Tobo et al., 2013), but
is still above the INP concentrations models suggest influence the properties of certain cloud types, such as high
latitude cold-sector clouds (Vergara-Temprado et al., 2018). The alternative approach is therefore to increase the
number of particles within each aliquot of water.  In principle, increasing the number of particles per droplet, and
therefore the surface area of nucleator, per droplet will increase the sensitivity of the experiment to rarer INP. This
enables quantification of lower INP concentrations. To increase the number of aerosol particles per volume of
liquid the time period over which an atmospheric sample is collected can be extended, but in doing so temporal
resolution would be lost. A method of increasing the sensitivity of an immersion mode technique is to increase
the volume of the collected suspension used in each aliquot, while maintaining the concentration of particles per
unit volume. This increases the number of particles per aliquot of liquid and therefore makes it more likely that
rarer INP will be detected.  The use of larger volume droplet suspensions has been exploited in the past (e.g. Bigg,
1953; Vali, 1971), and has been the strategy employed in the development of some recent instruments (e.g. Beall
et al., 2017; Conen et al., 2012; Du et al., 2017; Stopelli et al., 2014). These large volume assays capture the rarer,
more active INP but often miss the more abundant but less active INP. Hence they should ideally be used alongside
a smaller droplet instrument to generate complimentary datasets.
While many instruments use optical cameras to detect freezing events (Beall et al., 2017; Budke and Koop, 2015;
Häusler et al., 2018; Whale et al., 2015), some researchers have used techniques to detect the release of latent heat
associated with freezing.  For example differential scanning calorimetry (Marcolli et al. 2007; Pinti et al. 2012)
and infrared emissions (Zaragotas et al., 2016; Kunert et al. 2018) have been used. Zaragotas *et al.* (2016) use a
thermal camera to measure the temperature of individual aliquots within a 96 multiwell plate partially submerged
within an alcohol bath. This study investigated plant samples but suggested that the technique may be adapted for
atmospheric purposes. Very recently, Kunert *et al.* (2018) presented a similar set up to investigate biological
samples and collected aerosol. Unlike Zaragotas *et al.* (2016), Kunert *et al.* (2018) do not measure individual
droplet temperatures via infrared emissions but instead use multiple thermistors embedded in the sample holders
to infer temperature for the droplet array.

Here we propose a new technique, the IR Nucleation by Immersed Particle Instrument (IR-NIPI), for the detection
of INPs using large volumes of sample in the immersion mode. This instrument is part of the NIPI suite of
instruments that includes the μL-NIPI. When used together these devices allow measurements to be taken over a
very wide range of INP concentrations. The use of an infrared camera allows temperature measurements to be
made for individual droplets which helps reduce errors from horizontal gradients across the array of droplets and
the effect of heat release on the temperature of neighbouring wells. The unique design, in combination with a
Stirling engine-based chiller, is also compact making it ideal for field-based measurements and the use of
multiwell plates lends itself to future automation.
**2      Instrument Design**
**2.1      Operating principle**
Drop assays have been used extensively for ice nucleation experiments (e.g. Budke and Koop, 2015; Conen et al.,
2011; Garcia et al., 2012; Knopf and Forrester, 2011; Stopelli et al., 2014; Vali, 1995, 1971; Whale et al., 2015).
This is partly due to their simplicity compared to other techniques but also the ability to scale the amount of
nucleator with droplet size. In brief, aqueous suspensions are prepared and droplets of a well quantified size are
placed onto a substrate or immersed in oil. These droplets tend to be monodispersed but polydispersed experiments
are also possible (Murray et al., 2011; Vali, 1971). The system is then cooled and the fraction of droplets frozen
is recorded. The cooling can be conducted at a constant rate or with a stepped rate to hold the droplets at a specified
temperature for a period of time (i.e. isothermally) to explore the time dependence aspect of ice nucleation
(Herbert et al., 2014; Sear, 2014; Vali, 1994). The droplets are monitored and the freezing temperature of each
droplet is recorded. The fraction of the droplet population frozen throughout the explored temperature range can
then be determined, from which the ice-nucleating active site density or INP concentration can be derived (Vali
et al., 2015).
If the surface area of nucleant per droplet is known then it is common to express the nucleating ability of a material
as the density of active sites per unit surface area of nucleator, $n_s(T)$ (Connolly et al., 2009; DeMott, 1995). This
approach is based on the assumption specific sites on a nucleator's surface are responsible for ice formation. $n_s$ is
a cumulative term, i.e. as you move to cooler temperatures there are more features which may behave as an active
site as the energy barrier for ice formation decreases. $n_s(T)$ is calculated via equation (1).
$$n_s(T) = \frac{(-\ln(1-\frac{n(T)}{N}))}{A} \qquad (1)$$
Where $n(T)$ is the number of droplets frozen at a given temperature and $N$ is the total number of droplets. $A$ is the
surface area of nucleator within each droplet. Nucleation is a time-dependent stochastic process, but in
determining $n_s(T)$ the time dependence is neglected. This assumption is justified for many materials because the
diversity in activity of active sites leads to a much greater spread in freezing temperatures than the shift in freezing
temperatures associated with changes in cooling rate (Herbert et al., 2014; Vali, 2008).
**2.2    IR-NIPI design**
In brief an aqueous suspension is prepared and aliquots pipetted into the wells of a 96 multiwell plate which is
then placed on a temperature controlled stage. The cold stage and multiwell plate are enclosed by a Perspex cover
with an infrared camera mounted in its lid (Figure 1). The system is cooled at ~ -1 °C min$^{-1}$ until all droplets are
frozen (typically in a temperature range of 0 to -30 °C).  The temperature of the individual aliquots is monitored
using the IR camera which records a temperature map every 20 seconds. The temperature map is then analysed
with a semi-automated process using custom Python code to yield the freezing temperatures of individual wells.
The IR-NIPI has been designed around an Asymptote Ltd. VIA Freeze$^{TM}$ stirling cryocooler (Figure 1). The VIA
Freeze uses a Stirling engine to provide a convenient means of cooling without refrigerants or circulating liquids
and was primarily designed for use in cryopreservation applications. This chiller can achieve temperatures of -90
°C, hence it has more than enough cooling capacity for our application, and has sufficiently low power
requirements that allow it to be run from an automotive 12 V inverter. It also features an onboard datalogger and
internal computer with touch screen control. The VIA Freeze has been developed to accommodate multiwell
plates onto its aluminium cooling stage, which are ideal for large volume drop assays as they hold up to 200 µL
per aliquot (for the 96 well plates), allow the separation of droplets to reduce interference across cells and can be
supplied medically sterile. These multiwell plates have anywhere from 12-1536 wells (with maximum working
volumes of 6.9 mL to 2 μL, respectively). The most useful for this freezing assay are the 96 x 200 μL or 384 x 50
μL aliquot arrays and in the tests reported here 50 μL droplets (~2300 μm volume equivalent diameter) are used
in 96 well plates. We have used both polystyrene (Corning, CLS3788) and polypropylene plates (Greiner, M8060)
and observed no difference in freezing results between the two. To aid thermal contact between the multiwell
plate and the VIA Freeze a thermally conductive gap pad (RS components, 7073452) is located between the cold
plate and the multiwell plate, while a clamping system with screw threads applies mechanical pressure to the
multiwell plate to push the wells into the pad (Figure 1). A specially designed Perspex hood then encloses the
system to reduce contamination from the surroundings. The IR camera slots into the hood and captures an image
of the multiwell plate every 20 seconds (Figure 2a), storing the corresponding temperature data (Figure 2b) on a
removable memory card. The IR camera used here is a Fluke Ti9 Thermal Imager with 160 x 120 pixels. The
Stirling engine chiller is then set to cool down at 1.3 °C min$^{-1}$ which corresponds to 1 °C min$^{-1}$ ± 0.06 °C in the
wells due to a measured offset between the plate and aliquot temperatures. This ramp rate was selected based on
preliminary runs and justification for this cooling rate being equivalent to 1 °C min$^{-1}$ can be seen in the well
temperatures over time (Figure 2b). Once the system has initially cooled to 5 °C the temperature is held for 5 min
to allow time for the system to equilibrate. Following this the system continues to ramp down in temperature while
recording IR heat maps of the multiwell plate.
In order to determine the temperature of individual wells, the analysis code locates a pixel centred in the middle
of each well, reporting this temperature as the well temperature. Profiles of temperature versus time are shown in
Figure 2b and c. The freezing temperature of each individual well is determined by comparing each temperature
reading, for a certain well, with the temperature recorded 20 seconds prior. If the temperature reading increases
by more than 2 °C this is recorded as a freezing event (Figure 2c). The 2 °C threshold occasionally needs to be
optimised to capture freezing events while eliminating the detection of false freezing events. For example, samples
that freeze above -3 °C are more difficult to detect because there is less heat released on initial freezing and
crystallisation happens over a longer period of time (see section 2.4). Manual inspection is required in this
temperature regime and the 2 °C threshold adjusted accordingly. The code then prints out the number of events
recorded, along with a time vs temperature plot (Figure 2b) and the corresponding event temperatures for the user
to quality control check and then exports the data as a '.csv' file.
The whole process from sample preparation to final analysis takes approximately 1 hour. In order to achieve
higher throughput of samples, albeit with a reduced number of replicates, multiple samples and internal blanks
can be placed within one multiwell plate. For example, when performing dilutions we might run 12 wells as a
handling blank and three lots of 28 wells that contain three different sample dilutions. This not only speeds up
analysis, it also reduces the effect of any time-dependent aging processes such as the rapid deactivation of an
albite sample suspended in water observed by Harrison et al. (2016).

**2.3     Temperature measurements with an Infrared camera**
By using an IR camera to view the thermal emission of each individual well of suspension we are able to obtain
temperatures associated with individual wells. This contrasts with the approaches adopted in other experiments
where the temperature is recorded and assumed to be representative for all droplets, for example when employing
a cold stage housing an embedded thermocouple whose reading is used to represent the temperature of the droplet
array. We note that in our system there was a lateral gradient across the entire multiwell plate in the IR-NIPI of
up to 6 °C (in extreme cases). This is likely due to there not being an even thermal contact of the multiwell plate
with the underlying cold plate. The typical gradient was 4 °C, hence temperature measurements of the individual
wells was necessary.
**2.4     Temperature calibration**
The IR camera we use was quoted for use between -20 to +250 °C with an uncertainty of ±5 °C and is intended
for use in a wide range of applications with a range of materials of different emissivity. In our application, we
only need to measure the temperature of one material, water, over a relatively narrow range of temperatures, hence
we perform a calibration for our specific experimental setup. Our calibration is based on the fact that when an
aliquot of water in a multiwell plate freezes, the released latent heat raises the temperature of the aliquot to the
ice-water equilibrium temperature (0 °C when the water activity of the sample is ~1, as it is in these experiments).
This is illustrated in Figure 2c which shows the phases of crystallisation that the aliquots go through. Initially, the
crystal growth is rapid with a rapid release of latent heat and a corresponding rise in temperature of the aliquot
within the 20 s time between frames. Visual inspection of the live screen display of the IR camera revealed that
the temperature reached a maximum within 1 s. The temperature of an ice-water mixture will necessarily be 0
°C, hence the aliquot cannot warm above 0 °C and the temperature will remain at 0 °C until all of the water has
frozen and no more heat is evolved. The rate of crystallisation in this regime is determined by the loss of heat to
the surroundings, in this case the cold stage, as well as to the surrounding droplets and the multiwell plate. This
stage of crystallisation takes longer at higher freezing temperatures where the temperature differential between
the cold stage and the aliquot is smaller.  Hence, freezing when nucleation takes place at -12 °C takes around 100
s, whereas when nucleation takes place at -20 °C freezing takes around 20-40 s.  Once all of the water has frozen
the temperature of the aliquot decreases rapidly back to that of the multiwell plate within 20-40 s. The fact that
the aliquots spend 10s of seconds at 0 °C provides a very useful calibration point for each individual well. In the
following we describe a novel method for calibrating the IR temperature measurements that takes advantage of
this process and proceed to justify this approach.
Using the analysis code, an event is identified and recorded. The code then reads the temperature of the frame
directly after this freezing event and calculates the difference of this value compared to 0 °C to give an offset
correction value, i.e. if the frame after freezing read 2 °C then the correction factor for this well would be -2 °C.
This offset value is then subtracted from all of the temperature recordings for that specific well. The average
correction value calculated for the IR camera via this method is -1.9 °C with a standard deviation ±0.5 °C. It
should be noted that one of the limitations of the setup used by Zaragotas *et al.* (2016) was that the IR camera was
calibrated only once by the factory, however our calibration method mitigates this limitation.
A standard freezing experiment was then performed and the thermocouple data was contrasted to that of the IR
camera which was calibrated using the above method (Figure 3). The comparison in Fig. 3a shows that the IR and
thermocouple temperature were in excellent agreement and this is also readily seen in residuals plotted in Fig 3b.
The scatter around the zero line in the residual plot is ± 0.9 °C (two standard deviations) in the regime after the
equilibrium step at +5 °C and before the first freezing event. We used this value as an estimate of the temperature
uncertainty associated with the IR technique generally. We did not use data in the red and blue shaded areas of
Fig. 3b to calculate this uncertainty. The temperature readings in the red shaded area were discarded as they had
not been held at +5 °C for five minutes to equilibrate. Temperature readings after the initial freezing event were
also discarded as thermal conductivity of ice is different to that of water and neighbouring wells release heat on
freezing which influence the temperature of surrounding wells.
We also tested the IR temperature measurement using T type thermocouples distributed in specific wells of a
polypropylene multiwell plate. The IR camera could not take an accurate reading of wells that had a thermocouple
placed inside them, therefore neighbouring unfrozen wells were assumed to be representative of each other (see
inset in Figure 4). As mentioned above, there is a gradient across the entire plate (Fig. 2A) and so a series of

preliminary experiments were undertaken to find suitable placement locations for the thermocouples in which the surrounding wells displayed similar temperature readings compared to one another. The thermocouples were placed in the base of the well along with 50 μL of Milli-Q grade water and four surrounding well temperatures were measured using the IR system. The thermocouple wire crossed one of the four IR measured wells and so only three wells adjacent to the thermocouple monitored well were used for comparison.

A total of six IR measurements were recorded with the corresponding thermocouple readings over a series of experiments spanning a temperature range of 20 °C to –25 °C. An example of a thermocouple measurement contrasted to three IR measurements can be seen in Figure 4a. The residual temperatures for all six thermocouple temperatures are also shown (Figure 4b). The IR temperature uncertainty derived from the aluminium well experiment is also plotted and shows that the multiwell temperature data is consistent with an uncertainty of ± 0.9 °C.

## 3  Test experiments and analysis

### 3.1  Control experiments

In larger volume freezing assays (10s of microliters) it is extremely challenging to remove all background INPs from the water and substrates, hence freezing is typically observed at temperatures well above what one would expect for homogenous freezing (Koop and Murray, 2016). Homogeneous nucleation is expected to result in 50 % of 50 μL droplets freezing at around -33 °C, whereas 50 % of the Milli-Q water droplets froze around -22 °C in our control experiments (Figure 5). Filtering of the Milli-Q water to 0.2 μm reduced the temperature at which pure water droplets froze by 2-3 °C. Sartorius Ministart, non-pyrogenic, single use filters were used for this (product code 17597-K). Blanks were run initially with entire 96 well plates and then 12 wells of each experiment thereafter were allocated for an internal blank when testing samples of INPs (i.e. 12 aliquots of Milli-Q water and 84 aliquots of sample suspension). Comparison of fraction frozen curves for typical IR-NIPI blanks with curves obtained for droplets containing various ice-nucleating materials (discussed below) show that there is a clear heterogeneous freezing signal (Figure 5). We hope to improve the baseline in the future, through further improvements in the cleanliness of the system (Polen et al., 2018), but for the purpose of these experiments the nucleants tested were active at sufficiently warm temperatures to be well above the baseline.

### 3.3  Feldspar chips

To further test the temperature readings from the IR-NIPI instrument a set of experiments was performed where
each droplet contained a single ~100 μm sized grain of K-feldspar in both the IR-NIPI and contrasted to the
standard μL-NIPI employing 1 μl droplets. The μl-NIPI is a well-established technique (Whale et al., 2015) which
compares well with other similar instruments (DeMott et al., 2018; Hiranuma et al., 2015). The purpose of this
experiment was to have the same amount of material per droplet in each experiment and to have the material at
the base of the droplet so that the results from the two instruments could be directly compared. In doing so we
could investigate the extent to which the gradient within the 50 μL wells might be a problem. This experiment
was adapted from the procedure described by Whale *et al.* (2018) and involved taking K-feldspar rich chips from
a bulk rock of pegmatite and selecting individual grains (pegmatite is an igneous intrusive rock rich in K-feldspar
with large grain sizes often being larger than 2.5 cm and hence easy to separate). This material was chosen because
K-feldspar is known to exhibit excellent ice-nucleating properties (Atkinson et al., 2013; Harrison et al., 2016;
Peckhaus et al., 2016). A total of 19 grains were ~100 μm in diameter were separated by eye, assigned a number
and their position tracked through the course of each experiment. The same feldspar chips were tested in both the
μL-NIPI and the IR-NIPI. For the IR-NIPI experiments single grains of feldspar were placed into the bottom of a
multiwell plate and 50 μL of Milli-Q water was then pipetted into each well. The experiment was then carried out
as normal and the freezing temperatures of the wells were recorded. The grains were then used in the μL-NIPI
experiment by placing the grains onto a glass cover slip atop a cold plate and pipetting a single 1 μL droplet onto
each grain, before carrying out a standard μL-NIPI experiment. Briefly, the temperature of the cold plate was
reduced at 1 °C min$^{-1}$ and the temperature of the droplet freezing events recorded via a camera. The resulting
fraction frozen plot for this experiment can be seen in
Figure 6a and the corresponding correlation plot is shown in Figure 6b. The two instruments yielded similar
fraction frozen curves and the individual feldspar grains nucleated ice at a similar temperature in both experiments.
The correlation plot in Figure 6b shows that the freezing temperatures of a single grain were not identical in the
two experiments, which is consistent with the stochastic nature of nucleation at active sites that have a
characteristic freezing temperature (Vali, 2014, 2008). The agreement between the two instruments suggests that
the temperature measurement and calibration of the IR-NIPI were robust and that there is no major temperature
gradient within the aliquots in the multiwell plates.

**3.4    NX-illite**
The mineral dust NX-illite was chosen as a test sample as it has been used in an extensive intercomparison study
(Hiranuma et al. 2015) and contains some common components which are found in atmospheric mineral dusts
(Broadley et al., 2012). NX-illite was taken from the same batch as that used by the Leeds group in the Hiranuma
*et al.* (2015) intercomparison and no further processing of the material was carried out. Aqueous suspensions of
the sample were prepared by weighing a known amount of material and suspending it in a corresponding volume
of water to make up a weight percent suspension (i.e. 0.1 g of mineral in 9.9 g of water to yield a 1 wt%
suspension). NX-illite concentrations of 0.01, 0.1 and 1 wt% were prepared in this manner, and in each case a
Teflon-coated magnetic stirrer bar was used to keep the particles suspended whilst the sample was pipetted into
the wells of the multiwell plate. Each concentration of NX-illite was tested using the IR-NIPI and the resultant
fraction frozen curves are shown in Fig. 5.
By employing a suspension of known concentration and composed of a material with a known specific surface
area, the surface area of nucleator per droplet can be calculated and used alongside the fraction frozen curves to
determine $n_s(T)$, as described in equation (1). The $n_s(T)$ derived from IR-NIPI with 0.01, 0.1 and 1 wt% NX-illite
are shown in
Figure 7a. They are in good agreement with one another with lower wt% suspensions yielding data at lower
temperatures and higher $n_s(T)$ values, as expected. The few data points from the 0.01wt% NX-illite run 2 which
appear as outliers may indicate that the particles were not evenly distributed throughout the droplets. Further to
this a freeze thaw experiment of 0.1wt% suspension was conducted where the sample was frozen once, thawed
and then frozen again (see Figure 8). The agreement between the two runs show that the material did not alter on
freezing.
The values of $n_s(T)$ for NX-illite derived from 0.01-1 wt% suspensions are shown in
Figure 7a together with the literature data for this material in
Figure 7b. This material has also been investigated by Beall *et al.* (2017) using an instrument that also uses 50μL
droplets: the Automated Ice Spectrometer (AIS).   The results of Beall *et al.* (2017) are therefore directly
comparable to the results from the IR-NIPI. All of the wet suspension techniques have been grouped together in
black in Fig. 7b, apart from the AIS data shown in green and the IR-NIPI data in red. Both the IR-NIPI and AIS
data are in good agreement with one another. It can be seen that the larger volume assays (IR-NIPI and AIS) give
results towards the upper spread of literature data but are still consistent with other results (

Figure 7b). Dry dispersed techniques have also been plotted as unfilled blue squares in Fig. 7b, but none of these techniques are sensitive in the range of $n_s(T)$ seen by the large droplet instruments. The new data from the IR-NIPI has extended the dataset for NX-illite to warmer temperatures than in previous measurements, illustrating the utility of the technique.

It should be noted that in preliminary experiments some discrepancies between dilutions of NX-illite were observed which highlighted the importance of accurately making up suspensions. In the following we note some issues that had to be solved. In some initial experiments the dilutions of a suspension would yield a higher than expected $n_s(T)$. On further investigation this issue was resolved via gravimetrically weighing suspensions (i.e. preparing a known mass of a sample in a known mass of water) rather than diluting a bulk stock suspension. Further to this great care was taken when sampling from the bulk NX-illite sample as to make sure no bias was introduced when selecting material since a powder can separate on the basis of grain size. This was avoided by shaking the container horizontally and selecting material from the centre of the bulk sample. Magnetic stirrer bars were used to keep particles suspended but when it came to collecting the suspension using a pipette the suspension was taken from the magnetic stirrer plate to stop the vortex within the vial. As the suspension was not stirring for a short period of time it meant that particles did not have time to fallout of suspension and there was no longer a vortex created by the stirrer bar which could bias particle distribution when sampling. The above emphasises the importance of selecting samples in a reproducible way and may explain some of the variability between the literature data seen in Fig. 7b.

## 3.4    Atmospheric aerosol sample

In order to demonstrate the utility of this approach for atmospheric aerosol samples, a filter sample was collected in Leeds as part of a field campaign held on the evening of the 5[th] November.  A sample of atmospheric aerosol was collected using a Mesa PQ100 air sampler for 100 min. An inlet head with an upper cut-off of 10μm was utilised and air was sampled at 16.7 L min[-1] on to a 0.4 μm polycarbonate track-etched Whatman filter, with a total of 1670 L of air sampled. The filter was then placed into 6 mL of Milli-Q water and vortexed for 5 min to wash the particles from the filter and into suspension.

The aqueous sample was then analysed on the IR-NIPI and μL-NIPI (Whale et al., 2015). The concentration of INPs per litre of air, $[INP]_T$, was subsequently calculated using equation (2) (DeMott et al., 2016).

$$[INP]_T = -\ln\left(\frac{Nu(T)}{N}\right)\left(\frac{V_w}{V_a V_s}\right) \qquad\qquad (2)$$
Where $N_u$(T) is the number of unfrozen droplets at a given temperature, $N$ is the total number of droplets, $V_w$ is
the volume of wash water, $V_a$ is the volume of an aliquot and $V_s$ is the volume of air sampled.
The resulting INP concentrations from the combination of these two instruments spanned four orders of magnitude
and covered a temperature range of 20 °C (see Figure 9). The data from both instruments was in good agreement
and yielded complementary information. This illustrates how the IR-NIPI can be used to extend the measurements
of INP concentrations to higher temperatures and lower INP concentrations. Since high-resolution regional
modelling of the effect of INP on shallow clouds suggests that 0.1 to 1 INP L$^{-1}$ is a critical concentration and much
lower concentrations still impact clouds (Vergara-Temprado et al., 2018), measurements with IR-NIPI will be
extremely useful, particularly in environments with low INP concentrations.
**4        Summary and conclusions**
The IR-NIPI technique is a novel approach to measuring freezing events in immersion mode nucleation studies.
We demonstrate that IR thermometry is a sound method for determining the freezing temperature of 50 μL water
droplets in multiwell plates. This method overcomes potential distorting influences such as thermal gradients
across the plate, the effect of freezing wells warming surrounding wells and poor thermal contact to the underlying
cold plate. A freezing event is detected as a sharp rise in freezing temperature to the equilibrium melting point
and a novel calibration method has been proposed which relies on the return of water droplets to the equilibrium
melting temperature of water, 0 °C, after initial freezing. This gives an individual calibration for every run and
every well. When comparing this calibration technique to thermocouple readings the data is consistent to within
±0.9 °C. The use of this calibration method is further supported when looking at experiments using single grains
of feldspar, with the results being consistent with those of the established μL-NIPI instrument that employs 1μL
droplets on a cold stage. Results for the ice nucleating ability of NX-illite with the IR-NIPI, a mineral dust which
has been the subject of an extensive inter-comparison, are consistent with literature measurements. In particular,
the IR-NIPI is in good agreement with another well-characterised large droplet instrument (AIS) (Beall et al.,
2017). However, it is unclear why both of these large volume instruments produce $n_s$ results at the high end of the
range of $n_s$ values reported previously. The utility of IR-NIPI for the analysis of atmospheric samples was also
demonstrated by collecting and analysing an aerosol sample from the city of Leeds, England. The sample was
analysed simultaneously with the μL-NIPI instrument.  Results from the two instruments were in good agreement
with one another. The IR-NIPI instrument extended the range of INP concentrations shown by the μL-NIPI by
two orders of magnitude, covering a regime critical for cloud formation with a modest sampling time of just 100
mins at 16.67 L min$^{-1}$.

*Data availability:* Data collected for the IR-NIPI temperature measurements, blank runs, NX-illite experiments
and field collected sample are available at http://dx.doi.org/10.5285/858a4b439d7d4466b82ea5215614f135.
*Acknowledgements:* We would like to take the opportunity to thank David Harrison for the assembly of the IR
camera automatic shutter trigger. We would also like to thank Antony Windross and Stephen Burgess for
construction of the camera housing and multiwell plate mount. The authors acknowledge the European Research
Council (MarineIce: 648661, CryoProtect: 713664 and IceControl: 632272), and the Natural Environment
Research Council (NE/M010473/1) for funding this research.

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

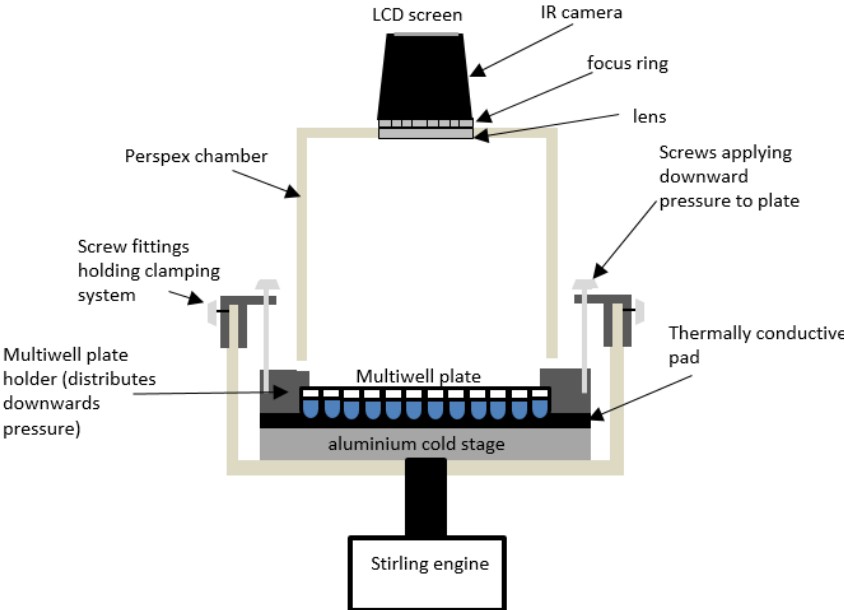


**Figure 1**. Schematic diagram of the IR-NIPI system (not to scale). The IR camera is positioned above the multiwell plate and monitors the
freezing events as the cold stage cools.



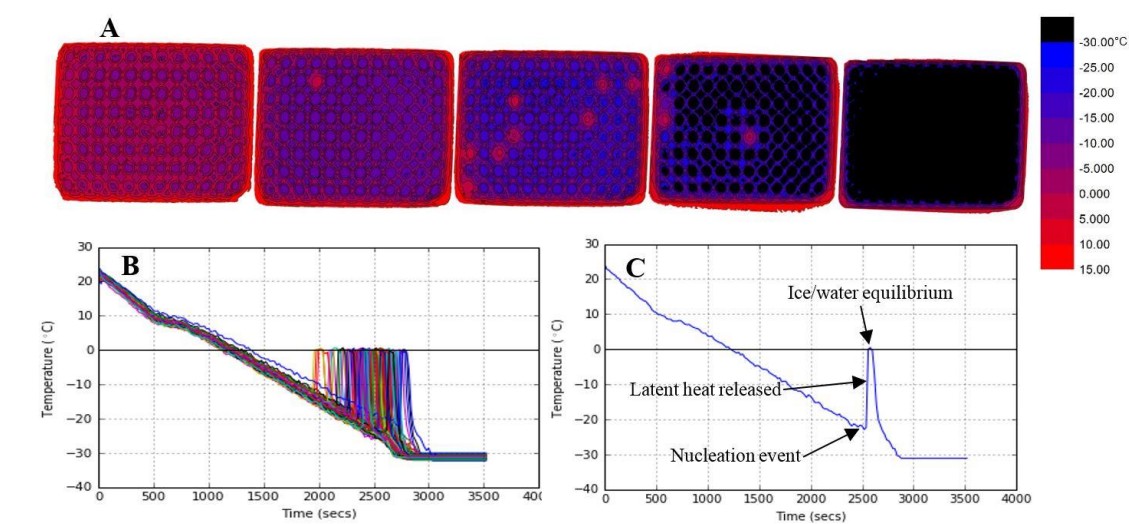

**553**

**554**

**555**

**556**

**557**

**558**

**559**

**560**

**561**

**Figure 2.** Illustration of the use of the IR camera to measure temperature and freezing events. (**A**) A sequence of colour maps taken during the course of an experiment. The leftmost image shows the start of an experiment with all droplets unfrozen, moving to all droplets frozen in the right most image. Warmer temperatures are represented in red, transitioning to blue for colder temperatures and finally black at -30 °C and below. Freezing events in individual wells are evident when they warm up to 0 °C. (**B**) An example of the output of an experiment with the temperature of each of the 96 wells plotted against time. The sharp increases in temperature are related to ice formation. The cooling rate was 1 °C min$^{-1}$. The calibration described in section 2.2 was applied here. Note that one well had a higher temperature than the others, likely due to poor thermal contact with the aluminium substrate. By using IR thermometry to measure the temperature of each well individually such variability is accounted for. (**C**) Plot of temperature vs time for a single well within a multiwell plate containing a 50 μL aliquot of water.

**570**

**571**

**572**

**573**

**574**

**575**

**576**

**577**

**578**

**579**

**580**

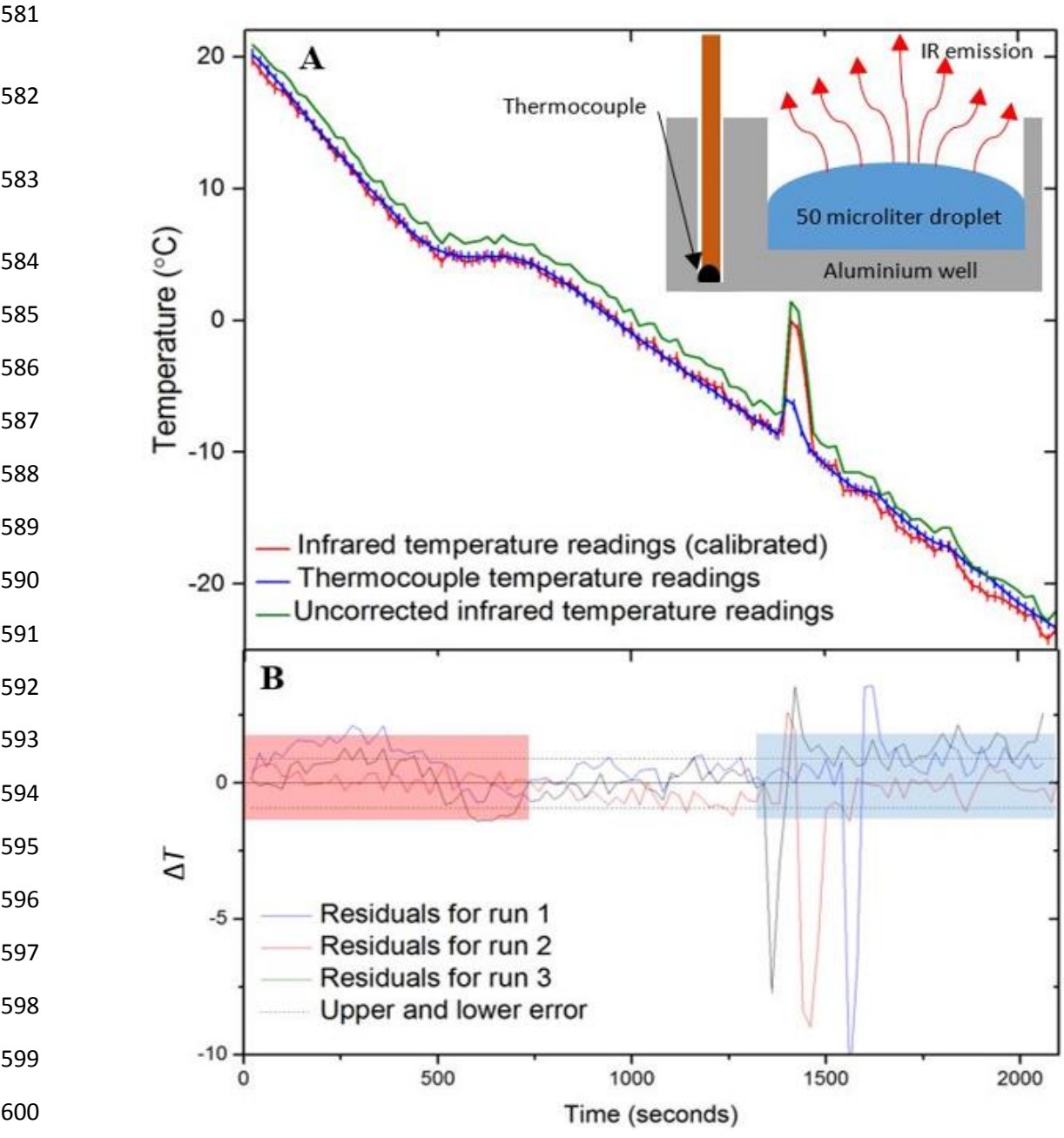

**Figure 3.** Tests of the IR temperature measurement using aluminium wells with embedded thermocouples. **(A)** The temperature measurement from a thermocouple (shown in blue) placed within an aluminium well vs infrared measurements taken using the IR camera. Uncorrected IR data is shown in green, whilst corrected IR data following the calibration described in section 2.2 is shown in red. Inset is a schematic of the experimental setup. **(B)** The difference in temperature (residual; $\Delta T$) between the thermocouple readings for three aluminium wells and the corresponding IR data ($\Delta T = T_{IR} - T_{Thermocouple}$). The negative spikes are a result of the IR camera directly reading the water temperature as it is heated by ice formation whereas the thermocouple measurement is reading the temperature of the aluminium well which is less affected by the latent heat release. The estimated error in temperature for the IR camera of $\pm 0.9°C$ is indicated with dashed lines. The range over which freezing occurs is highlighted with a blue rectangle as this is where the thermal properties of ice and the initiation of heat release affect the temperature readings. Highlighted in red is the section of data before the well has equilibrated and so the IR camera is likely reading a warmer surface temperature than the thermocouple. See text for discussion.

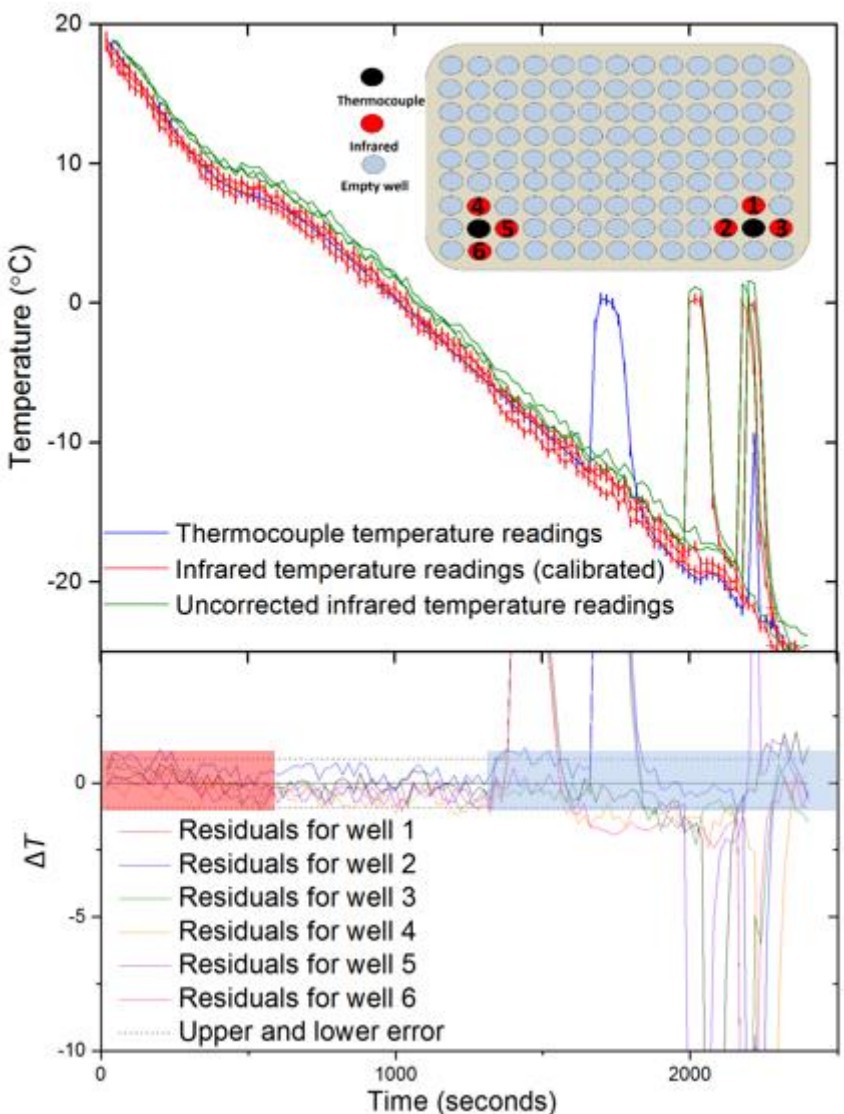

613

**Figure 4.** Tests of the IR temperature measurements using thermocouples positioned in a multiwall plate. **(A)** The temperature measurement from a thermocouple placed within a polyethylene well verses three IR measurements of surrounding wells corrected using the calibration described in section 2.2. The uncorrected IR data can be seen in green, with the corrected IR data in red and the thermocouple readings in blue. A diagram of the wells within a 96 well plate chosen for the comparison of IR and thermocouple measurements is displayed as an inset. The numbers of the wells correspond to the residuals in part B. Red wells represent the wells measured with the infrared camera and black wells represent those measured with thermocouples. It should be noted that one of the four surrounding IR well temperature readings was discarded from each experiment as the thermocouple wire impeded the temperature measurement. Note that the freezing events at ~2000 s appear to cause some heating in the adjacent well. **(B)** Plot of the difference in temperature between the thermocouple readings for two wells and six corresponding wells measured with the IR camera as in Figure 3. The estimated error in temperature for the IR camera is indicated with dashed lines (±0.9℃). The range of freezing is highlighted in blue as this is where the thermal properties of ice and the initiation of heat release will affect the temperature readings. Highlighted in red is the section of data before the well had equilibrated and so the IR camera was likely reading a warmer surface temperature than the thermocouple.

626

627

628

629

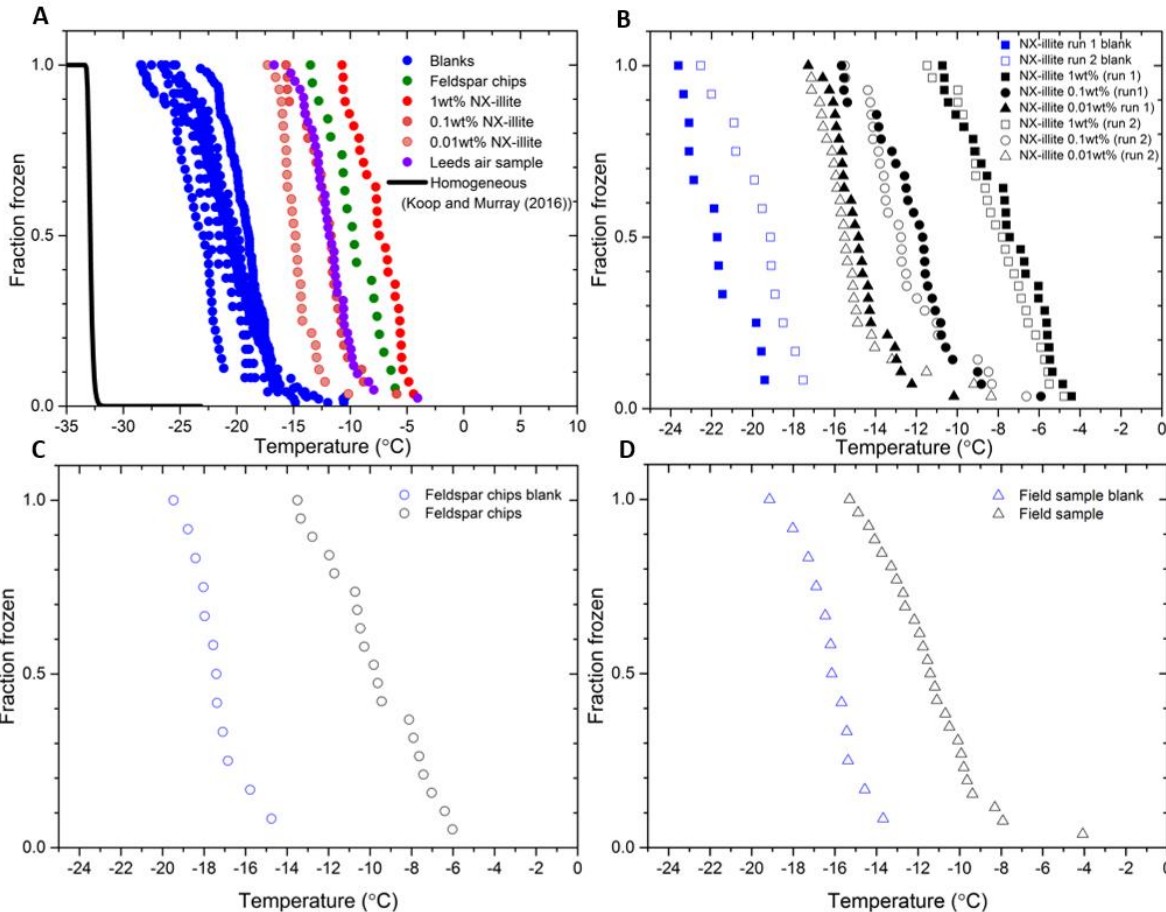

630

**Figure 5.** The fraction of droplets frozen as a function of temperature on cooling for a range of samples. (**A**) The fraction frozen curves for
the IR-NIPI experiment showing standard blank runs and the sample runs from this study. The homogeneous freezing of water as predicted
with the Koop and Murray (2016) parameterisation is also shown in black. (**B**) Fraction frozen plot for the internal blanks versus the
corresponding NX-illite runs. (**C**) Fraction frozen plot for the internal blank versus the corresponding feldspar chips run. (**D**) Fraction frozen
plot for the internal blank versus the corresponding field sample run.





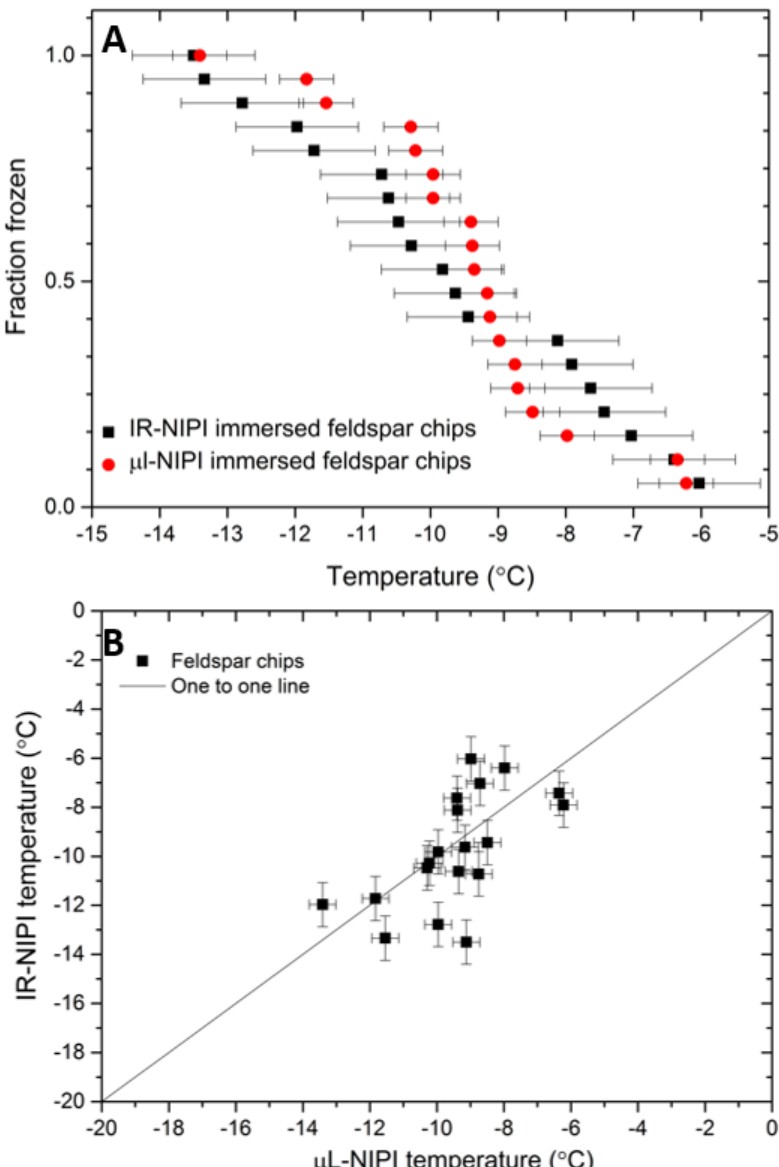



**Figure 6.** Comparison of nucleation induced by feldspar chips in the IR-NIPI and μL-NIPI instruments. **(A)** The fraction frozen curves for
single feldspar particles per droplet in both the μL-NIPI (using 1 μL droplets) and IR-NIPI (using 50 μL droplets) experiments. The error bars
indicate the error in temperature measurement on both instruments, respectively. **(B)** Shows the freezing temperature for the individual feldspar
chips as measured by the IR-NIPI and μL-NIPI instruments.






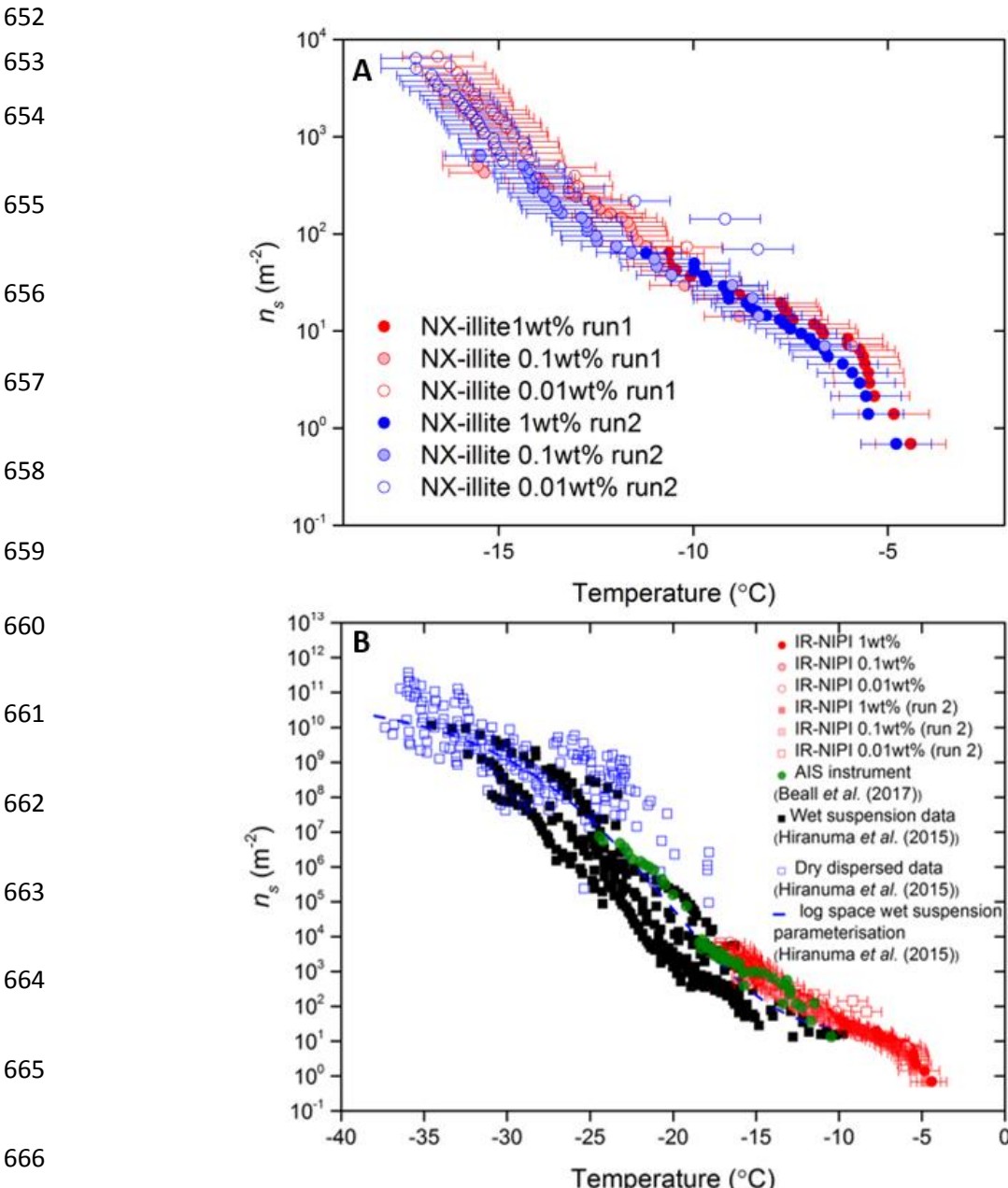

**Figure 7.** Active site densities, $n_s(T)$, for NX-illite. **(A)** The active site density for a dilution series of NX-illite run on the IR-NIPI instrument for a range of concentrations. The data for a repeat experiment is also shown. The error bars represent the temperature error of ±0.9°C. **(B)** The active site density for NX-illite from this study compared to literature data. Data from wet dispersed techniques are displayed in black with the IR-NIPI highlighted in red and Automated Ice Spectrometer (AIS) in green. Data from dry dispersed techniques are also plotted as hollow blue squares.




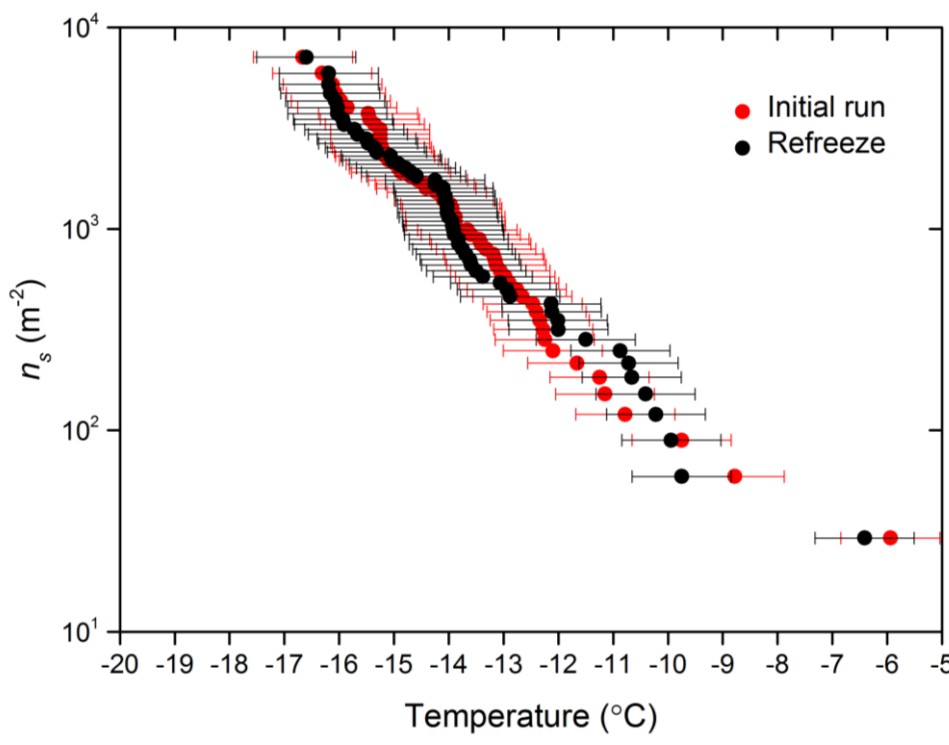


**Figure 8.** The active site density, $n_s(T)$, for a 0.01wt% NX-illite suspension and $n_s(T)$ from a corresponding second freezing run with the same
droplets after they had been thawed out and refrozen.








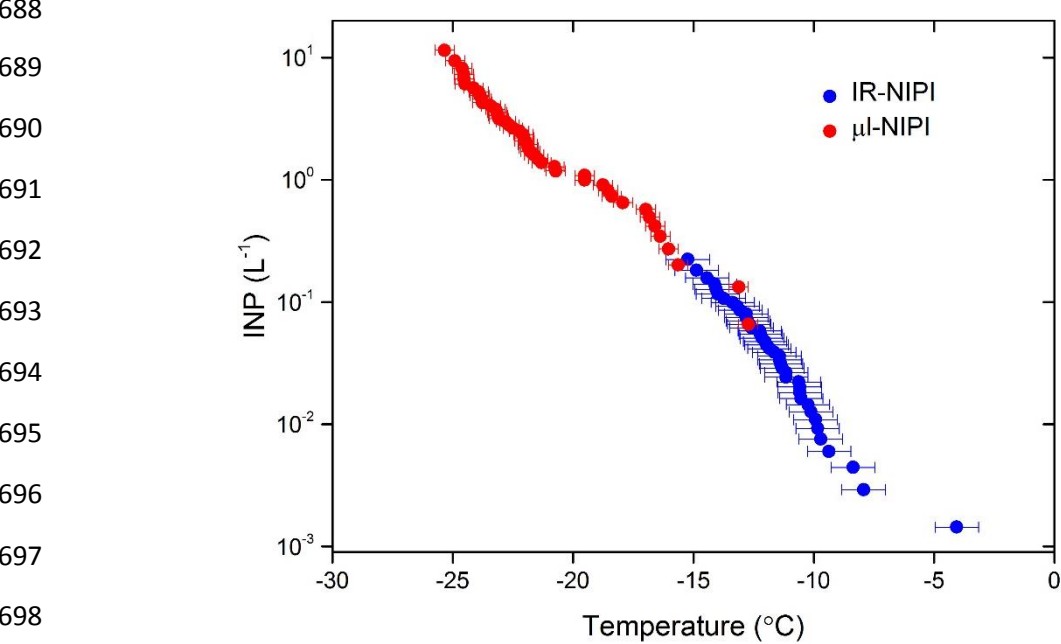

**Figure 9.** INP concentrations per litre of air samples on the 5th November 2017 in Leeds. Aerosol were collected onto filters for later extraction into water and analysis of the resulting suspensions with the IR-NIPI and µL-NIPI instruments.