# Peer review of "An instrument for quantifying heterogeneous ice nucleation in multiwell plates using infrared emissions to detect freezing"

_Atmospheric Measurement Techniques, 2018_

## Referee Comment (RC1) · Anonymous Referee #1 · 8 Jul 2018

Review of "An instrument for quantifying heterogeneous ice nucleation in multiwell plates using infrared emissions to detect freezing" by Harrison et al.

General comments: I support publication of this manuscript in AMT. The research aligns well with the scope of AMT. The reviewer finds the application of release of latent heat for detecting a freezing event in immersion mode ice spectrometer unique. The authors successfully present the applicability of their technique (IR-NIPI) to characterize immersion freezing efficiencies of three different forms of the sample (incl. chips, powder and ambient particles collected on the filter and scrubbed with water) at T > -22 °C. In particular, its applicability to the atmospheric sample seems promising

- the reviewer finds Figs. 8 and 9 nice and elegant. With further improvements in the temperature uncertainty ($\pm 0.9\ ^\circ$C is reported in the manuscript) and applicability in different droplet volumes (so, wider T coverage), IR-NIPI may become very versatile in the INP research (specifically for biological high-T INPs). The review has only minor (but important) comments.

Minor specific and technical comments: P1 L11-13: The main focus of the presented work is on novel application of latent heat in immersion freezing spectrometry, and the reviewer finds the discussion of online vs. offline unnecessary (especially in the abstract). L12-13 is erroneous – some cloud simulation chambers can assess multiple Ts. The reviewer strongly suggests removing "While instruments . . . Hence,".

P2 L36-38: Reference suggestion - Hande, L. B., and Hoose, C.: Partitioning the primary ice formation modes in large eddy simulations of mixed-phase clouds, Atmos. Chem. Phys., 17, 14105–14118, 2017.

P2 L40: $\sim$1 L-1 at what temperature? Please clarify to the readers.

P2 L42-43: Plus developing realistic but computationally inexpensive parameterization is also a key to what is addressed here by the authors.

P2 L45: Quantitatively define "warmer temperatures" perhaps with specific reference(s). L51-53 implies -11 $^\circ$C as warmer temperatures?

P3 L61-63: What about the Arctic? Some discussions may benefit the paper.

P3 L71: Reference suggestion - Stopelli, E. et al.: Freezing nucleation apparatus puts new slant on study of biological ice nucleators in precipitation, Atmos. Meas. Tech., 7, 129–134, 2014.

P3 L71: The reviewer thinks the discussion of previous studies applying latent heat release as an asset for ice nucleation research will benefit the paper. Please consider include and discuss; e.g., Marcolli, C. et al.: Efficiency of immersion mode ice nucleation on surrogates of mineral dust, Atmos. Chem. Phys., 7, 5081-5091, 2007.

P4 L99: ns(T)

P5 L140-141: Very important statement – recap this point (sharp rise in T = +2 °C) in the abstract and/or conclusion section.

P6 L142-145: The authors may want to rephrase this part and explain the points more intuitively.

P6 L158-161: Very good. The authors do truly careful experiments/assessments.

P7 L179-182: The authors may want to extend this part and explain the points more intuitively.

P7 L189-192: The reviewer is curious if using different droplet volume can improve this uncertainty. The reviewer does not intend to ask any additional measurements (especially since ±0.9 °C uncertainty is well justified in L193-207), but doe the authors have any estimates of the maximum/minimum droplet volume that IR-NIPI can deal with?

P8 L214: Delete "see".

P11 L298-300: 16.7 L/min * 100 min = 1670 L. . . The authors might want to check their nINP (L-1) since they might have employed a wrong Vs (Eqn. 2).

P11 L304: Was a dilution used to prepare suspensions for the ambient sample analysis? If not, no worries. But, if yes, the dilution factor is missed in Eqn. 2.

Fig. 7A: There seems some outliers within this T-ns(T) scale (i.e., 0.01 wt% run 2). What is responsible for them? Perhaps, it is due to what is addressed in L280-293? Please clarify.

---

## Referee Comment (RC2) · Anonymous Referee #2 · 10 Jul 2018

The comment was uploaded in the form of a supplement:
https://www.atmos-meas-tech-discuss.net/amt-2018-177/amt-2018-177-RC2-supplement.pdf

---

## Referee Comment (RC3) · H. Grothe (Referee) · 12 Jul 2018

This is a well-written paper with an interesting experimental approach that fits perfectly well into the journal Atmospheric Measurement Techniques. The authors describe an instrument for quantifying heterogeneous ice nucleation: the InfraRed-Nucleation by Immersed Particles Instrument (IR-NIPI). They use multiwell plates and an infrared camera for detection of the freezing process. For comparison, they have investigated homogeneous ice nucleation of ultrapure water, and the heterogeneous freezing of two mineral dust samples. The manuscript should be published in AMT after major revisions.

[Figure]

General comments

In line 72, the authors claim "Here we propose a new technique,. . .". Unfortunately, this is not entirely true. A quick literature research shows that there are other instruments with very similar approaches. In particular, I would like to mention the set-ups of Zaragotas et al. and of Kunert et al. It is good scientific practice to search, to describe, and to discuss the findings of other scientists when presenting a new set-up. I expect that the authors make up the leeway in the revised version of the manuscript.

Concerning the set-up of Kunert et al., I could not find any peer-reviewed publication, but I have been the organizer of two ice workshops (Kunert 2016b, 2017b) and the convener of two EGU General Assembly sessions (Kunert 2016a, 2017a) and a speaker at the INUIT Final Conference and 2nd Atmospheric Ice Nucleation Conference (Kunert 2018), where this research has been presented. At all these meetings, also the authors were present and in the case of the latter have even been the organizers. Therefore, the Twin-plate ice nucleation assay (TINA) with infrared detection by Kunert et al. is well-known to them and should be described in their manuscript for comparison with IR-NIPI.

In line 48, the authors list some droplet freezing assay experiments but the list is rather incomplete, e.g. Häusler 2018 is missing. I strongly recommend a table with all technical parameters of each experiment listed, e.g. number of observed volumes, volume of the droplets, homogeneous ice nucleation temperature, etc. Finally, for all experiments a discussion of the pros and cons in comparison to IR-NIPI should be added.

Instead, the authors compare only their own set-ups, i.e. $\mu$L-NIPI and IR-NIPI. However, the volume of the respective droplets is very different, $1\mu$L versus $50\mu$L, respectively. This is not only important for homogeneous ice nucleation, which shows strong volume dependence, but also is important for heterogeneous ice nucleation because larger volumes carry more INPs and the abundance of efficient INPs rises. The authors have discussed this only partly and a more elaborated discussion might be necessary.

In particular, I miss plots of the homogeneous freezing events and a detailed study of the freezing of single droplets (marked with numbers on a picture of the multiwell assay). I also recommend adding the diameter of the droplets to the volume to make the study more comparable to other studies.

The authors make the point that their set-up is more sensitive for low concentrations of INPs, which is particularly true for strong INPs. However, they don't mention the disadvantage of their set-up, which is that they cannot easily measure weak INPs. In the atmosphere, the number of strong INPs is extremely low, which makes $\mu$L-NIPI a valuable technique. However, often strong INPs are entirely missing and weak INPs will be much more abundant. Therefore, the authors should discuss the limitations of their set-up and should also show experiments at the detection threshold and should investigate proxies for weak INPs e.g. cellulose or soot. Also I miss biological INPs or proteins and polysaccharides been emitted by biological sources. Therefore, beside ns values of solid INPs also nm values of soluble INPs should be measured and discussed.

Specific comments

Where is the homogeneous freezing temperature (T50) of ultrapure water in your IR-NIPI set-up?

Have you quantified the effect of the walls of the multiwall plates on the freezing process?

Line 99: Also indicate the formula for nm and add respective water soluble samples.

Line 182: "standard deviation $\pm 0.5^\circ$C"

Line 190: "after the first equilibrium step at $+5^\circ$C"

How is the temperature uncertainty in the range between $-20^\circ$ and $-30^\circ$ C ?

You have only used ultrapure water for temperature calibration. How about other samples such as aqueous salt solutions, higher alcohols or alkanes?

Line 215: What kind of filter has been used for purification?

A figure, similar to that in fig. 7B, should be plotted also for feldspar samples including comparison data from other groups.

In figures 2, 3, 4, and 7 capital letters have been used in the graph but small letters have been used in the figure caption, respectively.

References

Häusler, T., Witek, L., Felgitsch, L., Hitzenberger, R., Grothe, H.: Freezing on a Chip - A New Approach to Determine Heterogeneous Ice Nucleation of Micrometer-Sized Water Droplets, Atmosphere 9 (4), 140, 2018.

Kunert, A. T., Scheel, J. F., Helleis, F., Klimach, T., Pöschl, U. and Fröhlich-Nowoisky, J.: New High-Performance Droplet Freezing Assay (HP-DFA) for the Analysis of Ice Nuclei with Complex Composition, in EGU General Assembly Conference Abstracts, vol. 18, pp. EPSC2016-6293., 2016a.

Kunert, A. T., Scheel, J. F., Helleis, F., Klimach, T., Pöschl, U. and Fröhlich-Nowoisky, J.: TINA: A New High-Performance Droplet Freezing Assay for the Analysis of Ice Nuclei with Complex Composition, in 4th Workshop - Microphysics of Ice Clouds., 2016b.

Kunert, A. T., Lamneck, M., Gurk, C., Helleis, F., Klimach, T., Scheel, J. F., Pöschl, U. and Fröhlich-Nowoisky, J.: TINA, a new fully automated high-performance droplet freezing assay coupled to a customized infrared detection system, in EGU General Assembly Conference Abstracts, vol. 19, p. 13571., 2017a.

Kunert, A. T., Lamneck, M., Gurk, C., Helleis, F., Klimach, T., Scheel, J. F., Pöschl, U. and Fröhlich-Nowoisky, J.: TINA, a new fully automated high-performance droplet freezing assay coupled to a customized infrared detection system, in 5th Workshop - Microphysics of Ice Clouds., 2017b.

Kunert, A. T., Lamneck, M., Helleis, F., Scheel, J. F., Pöschl, U. and Fröhlich-Nowoisky, J.: TINA: Twin-plate ice nucleation assay with infrared detection for high-throughput droplet freezing experiments, in INUIT Final Conference and 2nd Atmospheric Ice Nucleation Conference., 2018.

Zaragotas, D., Liolios, N. T. and Anastassopoulos, E.: Supercooling, ice nucleation and crystal growth: A systematic study in plant samples, Cryobiology, 72(3), 239–243, 2016. http://dx.doi.org/10.1016/j.cryobiol.2016.03.012

---

## Author Comment (AC1) · 6 Aug 2018

**Response to Referee #1**

We would like to thank the referee for their insightful comments and have responded below. The referee comments are highlighted in red with our responses in black.

Review of "An instrument for quantifying heterogeneous ice nucleation in multiwell plates using infrared emissions to detect freezing" by Harrison et al. General comments: I support publication of this manuscript in AMT. The research aligns well with the scope of AMT. The reviewer finds the application of release of latent heat for detecting a freezing event in immersion mode ice spectrometer unique. The authors successfully present the applicability of their technique (IR-NIPI) to characterize immersion freezing efficiencies of three different forms of the sample (incl. chips, powder and ambient particles collected on the filter and scrubbed with water) at T > -22 ∘C. In particular, its applicability to the atmospheric sample seems promising the reviewer finds Figs. 8 and 9 nice and elegant. With further improvements in the temperature uncertainty (±0.9 ∘C is reported in the manuscript) and applicability in different droplet volumes (so, wider T coverage), IR-NIPI may become very versatile in the INP research (specifically for biological high-T INPs). The review has only minor (but important) comments.

**Comments**

P1 L11-13: The main focus of the presented work is on novel application of latent heat in immersion freezing spectrometry, and the reviewer finds the discussion of online vs. offline unnecessary (especially in the abstract). L12-13 is erroneous – some cloud simulation chambers can assess multiple Ts. The reviewer strongly suggests removing "While instruments . . . Hence,".

Accepted, we agree and have removed this piece of text from the abstract.

P2 L36-38: Reference suggestion - Hande, L. B., and Hoose, C.: Partitioning the primary ice formation modes in large eddy simulations of mixed-phase clouds, Atmos. Chem. Phys., 17, 14105–14118, 2017.

Thank you. We have now added this reference.

P2 L40: ~1 L-1 at what temperature? Please clarify to the readers.

The study was of three case studies of cloud fields. The temperature at which this concentration was reached depended on the local INP spectrum. It is quite challenging to insert this information without distracting from the main point of the statement hence we would rather leave it as is.

P2 L42-43: Plus developing realistic but computationally inexpensive parameterization is also a key to what is addressed here by the authors.

We have changed the text to read "The ability to quantify INP spectra (INP concentrations as a function of temperature) and test the efficiency of proxy materials for ice-nucleating efficiency is invaluable for improving our understanding of cloud glaciation and developing computationally inexpensive parameterisations for atmospheric models."

P2 L45: Quantitatively define "warmer temperatures" perhaps with specific reference(s). L51-53 implies -11 ∘C as warmer temperatures?

We have altered the text to read "However it is not a trivial task, in part because INP concentrations are low ($<0.1 L^{-1}$) (DeMott et al., 2010) and the sites on the surfaces which cause nucleation at warm temperatures (Whale et al., 2017; Vali, 2014) are rare. "

P3 L61-63: What about the Arctic? Some discussions may benefit the paper.

We agree that the Arctic is similarly low in INP concentrations, hence expect a similar situation to the southern ocean. We have now discussed this. "This can be improved with aerosol concentrators (Prenni et al., 2013; Tobo et al., 2013), but is still above the INP concentrations models suggest influence the properties of certain cloud types, such as low/high latitude cold-sector clouds (Vergara-Temprado et al., 2018)."

P3 L71: Reference suggestion - Stopelli, E. et al.: Freezing nucleation apparatus puts new slant on study of biological ice nucleators in precipitation, Atmos. Meas. Tech., 7, 129–134, 2014.

This has been added. In addition we have added Conen et al. 2012.

P3 L71: The reviewer thinks the discussion of previous studies applying latent heat release as an asset for ice nucleation research will benefit the paper. Please consider include and discuss; e.g., Marcolli, C. et al.: Efficiency of immersion mode ice nucleation on surrogates of mineral dust, Atmos. Chem. Phys., 7, 5081-5091, 2007.

We agree a discussion of this paper should be made and have added the following text in a new section where we also discuss other instruments one of the other referees highlighted: "While many instruments use optical cameras to detect freezing events (Whale et al., 2015; Budke and Koop, 2015; Häusler et al., 2018; Beall et al., 2017), , some researchers have used techniques which detect the release of latent heat associated with freezing.. For example, differential scanning calorimetry (Marcolli et al. 2007; Pinti et al. 2012) and infrared emissions (Zaragotas et al., 2016; Kunert *et al*. 2018) have been used.

P4 L99: ns(T)

Done

P5 L140-141: Very important statement – recap this point (sharp rise in T = +2 ◦C) in the abstract and/or conclusion section.

We have recapped this in the conclusion "A freezing event is detected as a sharp rise in freezing temperature to the equilibrium melting point and a novel calibration method has been proposed which relies on the return of water droplets to the equilibrium melting temperature of water, 0°C, after initial freezing." The sharp rise in temperature is already referred to in the abstract.

P6 L142-145: The authors may want to rephrase this part and explain the points more intuitively

We have reworded this section so that the reader may better understand the points made. "The 2°C threshold occasionally needs to be optimised to capture freezing events while eliminating the detection of false freezing events. For example, samples that freeze above -3°C are more difficult to detect because there is less heat released on initial freezing and crystallisation happens over a longer period of time (see section 2.4). Manual inspection is required in this temperature regime and the 2°C threshold adjusted accordingly. "

P7 L179-182: The authors may want to extend this part and explain the points more intuitively

We have rewritten this section to explain the process more clearly "Using the analysis code, an event is identified and recorded. The code then reads the temperature of the frame directly after this freezing event and calculates the difference of this value compared to 0°C to give an offset correction value, i.e. if the frame after freezing read 2°C then the correction factor for this well would be -2°C. This offset value is then subtracted from all of the temperature recordings for that specific well. The average correction value calculated for the IR camera via this method is -1.9°C with a standard deviation ±0.5°C."

P7 L189-192: The reviewer is curious if using different droplet volume can improve this uncertainty. The reviewer does not intend to ask any additional measurements (especially since ±0.9 ◦C uncertainty is well justified in L193-207), but doe the authors have any estimates of the maximum/minimum droplet volume that IR-NIPI can deal with?

We have not completed any thorough experiments with different droplet volumes but the 96 multiwell plates can hold 200µL droplets. The IR-NIPI should have no problem monitoring these volumes although the gradients within the wells will become larger. Volumes below 50µL's maybe possible but would be starting to come close to the limits of the IR cameras resolution. If the IR cameras resolution were to be improved then smaller droplet sizes should be possible.

P8 L214: Delete "see".

Done

P11 L298-300: 16.7 L/min * 100 min = 1670 L. . . The authors might want to check their nINP (L-1) since they might have employed a wrong Vs (Eqn. 2).

Thank you for noticing this. This was a typo and has now been corrected.

P11 L304: Was a dilation used to prepare suspensions for the ambient sample analysis? If not, no worries. But, if yes, the dilution factor is missed in Eqn. 2.

No dilution was used.

Fig. 7A: There seems some outliers within this T-ns(T) scale (i.e., 0.01 wt% run 2). What is responsible for them? Perhaps, it is due to what is addressed in L280-293? Please clarify.

We believe this may be related to the issues discussed in L280-293. We have added a sentence to emphasise this in the results discussion of Fig. 7A. "The $n_s(T)$ values derived from IR-NIPI with 0.01, 0.1 and 1 wt% NX-illite are shown in 7a. They are in good agreement with one another with lower wt% suspensions yielding data at lower temperatures and higher $n_s(T)$ values, as expected. The few data points from the 0.01wt% NX-illite run 2 which appear as outliers may indicate that the particles were not evenly distributed throughout the droplets."

---

## Author Comment (AC2) · 6 Aug 2018

**Response to Referee #2**

We would like to thank the referee for their insightful comments and have responded below. The referee comments are highlighted in red with our responses in black.

This manuscript presents a new instrument to measure low concentrations of ice nucleating particles in the atmosphere. It relies on commercial multiwell plates to allot the samples extracted from filters. For cooling, it uses a Stirling engine chiller that can be easily operated in the field. Because the heat transfer between the chiller and the multiwell plate is not uniform, the temperature of each well is determined with an IR camera. The same camera is also used to detect freezing from the heat release. To retrieve the actual temperature of the wells, use is made of the temperature increase upon freezing. The plateau temperature reached during freezing is taken as 0°C and the whole temperature curve is adjusted to this reference value by applying an offset correction. Validation and calibration revealed a temperature uncertainty of ±0.9°C. The setup has been tested with K-feldspar chips, NX-illite and an aerosol sample taken from the City of Leeds. All these measurements show reasonable agreement with reference measurements from literature or performed with the µL-NIPI developed and available in the Leeds group. This manuscript is well suited for AMT and will be useful for other groups developing similar setups. For publication in AMT, the quality of presentation should be improved. Some descriptions remain vague and the figures are often not fully explained. In the introduction there is often only one reference given, while other relevant references to support the statements are lacking.

**Comments**

Lines 22 – 23: "and their temperature is determined by the ice-liquid equilibrium temperature…." This sentence should be improved because it is difficult to understand without having read through the manuscript.

We have amended the text to read as "In this paper we first present the calibration of the IR temperature measurement, which makes use of the fact that following ice nucleation aliquots of water warm to the ice-liquid equilibrium temperature (i.e. 0°C when water activity is ~1), which provides a point of calibration for each individual well in each experiment."

Line 46: literature referring to more recently developed CFDC's should be added.

References have been updated to: " There are several different methods of conducting ice nucleation experiments that include Continuous Flow Diffusion Chambers (CFDC's) e.g. (Salam et al., 2006; Rogers et al., 2001; Kanji and Abbatt, 2009; Stetzer et al., 2008; Garimella et al., 2016; Kohn et al., 2016),

Line 56: Vergara-Temprado et al. (2017) is not the best reference to support this statement. Other references need to be added, e.g. Kanji et al. (2017).

Kanji *et al.* (2017) added. Vergara-Temprado et al. 2017 was a modelling study in which field data was compiled and compared to the model. One conclusion from this study was that there was a lack of field data.

Line 61: Again, this statement needs to be based on more and more recent literature because different types of CFDC's are in use.

This has now been updated to include the following references "(Eidhammer et al., 2010; Al-Naimi and R., 1985; Prenni et al., 2009; DeMott et al., 2010)"

Line 61: the aerosol concentrator needs to be referenced.

Now referenced "This can be improved with aerosol concentrators (Prenni et al., 2013; Tobo et al., 2013),…"

Lines 64 – 66: "In principle, if the ice-nucleating properties of the aerosol particles in question are insensitive to mixing state, then increasing the amount of aerosol per droplet will scale with inverse proportionality to the INP concentration,…". This sentence needs to be improved. What is meant by "mixing state" in this context? What is meant by "increasing the amount of aerosol per droplet will scale with inverse proportionality to the INP concentration?" Do you really mean "aerosol per droplet" or not rather "particles per droplet"?

This has now been reworded to "In principle, increasing the number of particles per droplet, and therefore surface area of nucleator per droplet, will increase the sensitivity of the experiment to rarer INP. This enables quantification of lower INP concentrations."

Line 69: "volume of suspension used in each aliquot": this formulation needs to be improved. The volume of suspension can be easily increased by diluting the suspension, but I guess this is not what is meant. Moreover, if filter samples are collected, it needs to be explained why extracting the filter with less water or evaporation of some of the water used for extraction is not an option.

We are not ruling out other methods of increasing the number of particles per aliquot of water, there may be other alternatives.  We have focused on what we have done here. We have modified the section as follows: "The alternative approach is therefore to increase the number of particles within each aliquot of water.  In principle, increasing the number of particles per droplet, and therefore the surface area of nucleator per droplet, will increase the sensitivity of the experiment to rarer INP. This enables quantification of lower INP concentrations" To increase the number of aerosol particles per volume of liquid the time period over which an atmospheric sample is collected can be extended, but in doing so temporal resolution would be lost. A method of increasing the sensitivity of an immersion mode technique is to increase the volume of the collected suspension used in each aliquot, while maintaining the concentration of particles per unit volume. This increases the number of particles per aliquot of liquid and therefore makes it more likely that rarer INP will be detected.  The use of larger volume droplet suspensions has been exploited in the past e.g. (Vali, 1971; Bigg, 1953), and has been the strategy employed in the development of some recent instruments e.g. (Beall et al., 2017; Du et al., 2017; Stopelli et al., 2014; Conen et al., 2012).These large volume assays capture the rarer, more active INP but often miss the more abundant but less active INP. Hence they are frequently used alongside a smaller droplet instrument to achieve a complimentary dataset."

Line 131: Where does the difference in set cooling rate and realized cooling rate come from? Why is this difference constant?

This cooling rate comes from a series of preliminary experiments with the justification being seen in the ramp rates for the IR-NIPI seen in the figures. The difference is likely constant as it is a lag between the temperature transfer from the cooled aluminium plate to the multiwell plate.

Line 152: What is meant by "aging of a sample in water"?

It has been demonstrated that for some materials ice-nucleating ability will change with time spent in water, sometimes over short periods of time. We have altered the text to clarify this and added references that demonstrate this behaviour. "This not only speeds up analysis, it also reduces the effect of any time-dependent aging processes such as the rapid deactivation of an albite sample suspended in water observed by Harrison et al. (2016)."

Lines 155 – 156: How representative is the surface temperature for the temperature distribution within the wells? Heat transfer simulations of Beall et al. (2017) performed for their multiwell setup showed temperature gradients within the wells. Please comment.

Yes, there are likely to be gradients in the wells, but our feldspar chip experiment suggest the gradients within the wells are not biasing results beyond the quoted uncertainty. The IR camera gave similar results to those of the μl-NIPI instrument where gradients in the droplet are thought to be negligible. In the IR-NIPI the feldspar chips are at the base of the well and our IR camera is reading the water air interface, yet we still see good agreement with the μl-NIPI system (within the T uncertainty).

It should also be noted that we did experiments with both polystyrene and polypropylene plates. Polypropylene is an order of magnitude greater thermal conductor than polystyrene. Between the two plates we saw no difference in freezing temperatures of NX-illite and the calculated offsets between the two experiments were similar (within the $0.6^0$C standard deviation mentioned earlier). Hence, we again think that the gradients are minor in all cases.

We think this is discussed adequately in section 3.3.

Lines 164 – 165: The increase of the well temperature up to 0°C during freezing indicates a slow heat removal and limited thermal contact to the aluminum cold stage via the thermally conductive pad. From this, heating of adjacent wells is expected. Can you comment on this?

We expect to see a rise in temperature to equilibrium irrespective of thermal contact. The heat production during the initial crystal growth process is very rapid, much more rapid that the rate at which heat is dissipated for almost all droplet arrays even in µL droplets on an aluminium block. We do expect to see some heating of neighbouring wells but it is often minor and we do not have any quantitative data to describe the extent of this heating. However, you can see an example of the heating of neighbouring wells in Fig. 4A at ~2000 seconds where a well has frozen and the neighbouring wells increase in temperature. The use of individual temperature measurements of wells mitigates any impact this heating might have on freezing temperatures. In the caption we have added 'Note that the freezing events at ~2000s appear to cause some heating in the adjacent well.'

Line 168: how can you observe that the temperature maximum was reached within 1 s when you read out the IR camera only every 20 s? What do you mean by visual inspection?

We record an image every 20 seconds but the IR camera we use has a live screen display in which we can monitor the experiment. The text has been altered to reflect this "Visual inspection of the live screen display of the IR camera revealed that the temperature reached a maximum within 1 s. "

Line 173: Can you quantify the temperature differential between the cold stage and the aliquot?

We know that the cold plate is significantly colder than the water aliquots. This does not matter because we use the IR emissions to monitor temperature. As explained in the text the ramp rate is set to 1.3 °C min$^{-1}$ (which results in a cooling rate of 1 °C min$^{-1}$ in the aliquot) to account for a lag between the cold stage and aliquot. We do not think there is a need to quantify the offset of the coldstage and the aliquot any further as we measure temperature independently via the IR camera.

Lines 183 – 186: You mention several aluminum wells. How many did you test?

We used 16 in an experiment but only monitored 3 of them with thermocouples. This is now explained in the text: "We performed a number of experiments to test the IR temperature measurement calibrated using the above method. In the first instance we used highly conductive individual anodised aluminium wells for 50µL droplets. The temperatures of three of these wells were recorded independently using T type thermocouples embedded in the aluminium wells to give a representative temperature of the well and aliquot of water (see inset in **Error! Reference source not found.**)."

Lines 196 – 198: It would be helpful if you could add a figure illustrating the temperature gradient within the plate

The temperature gradient can be seen in figure 2a. This is now referenced in the text "As mentioned above there is a gradient across the entire plate (Fig. 2A)…"

Lines 204 – 205 and Fig. 4: From the scheme shown in the inset, results from 12 wells should have been measured, but in Fig. 4B results of only 6 wells are shown. Why? It would be helpful when you would label the wells shown in the inset with the numbers of the wells indicated in panel B.

Two wells measured with thermocouples were contrasted to three surrounding wells measured with the IR camera, so a total of six IR measurements were recorded. The schematic inset was to demonstrate that thermocouples were placed in the corners where the difference between the surrounding wells was at a minimum and that up to four of the surrounding wells could be used to compare to the thermocouple measured well. We accept the schematic is confusing and has been redrawn just to display the position of the wells observed.

Lines 215 – 216: By how many degrees was the freezing temperature reduced due to filtering of the Milli-Q water?

It looked to improve the baseline by ~2-3°C. This has been added to the text. "Filtering of the Milli-Q water to 0.2 µm reduced the temperature at which pure water droplets froze by 2-3°C. Sartorius Ministart, non-pyrogenic, single use filters were used for this (product code 17597-K)."

Section 3.3 and Fig. 6 (Feldspar chips): This section needs better explanations of the rationale and the execution of the experiment.

We have made section 3.3 clearer.

• You refer to Whale et al. (2015) for the setup but Whale et al. describes experiments with droplets containing a suspension of mineral particles and not with grains/chips. Please explain better the sample preparation of this study.

The reference in the text is for Whale *et al.* (2018) in which they describe this method. We later reference Whale *et al.* (2015) for the later stages of the experiment as the overall procedure of the experiment is explained better in this paper. This is clearer now in the revised section.

• How large were the grains/chips?

This is in the text as "each droplet contained a single ~100 µm sized grain of K-feldspar" but has been re-clarified later in the text as "A total of 19 grains were collected (~100 µm in size), assigned a number and their position tracked through the course of each experiment."

• How many droplets were deposited on a grain for the µL-NIPI experiment?

A single µL droplet was deposited on top of each grain. This has now been clarified in the revised text. "The grains were then used in the µL-NIPI experiment by placing the grains onto a glass cover slip atop a cold plate and pipetting a single 1 µL droplet onto each grain, before carrying out a standard µL-NIPI experiment."

• Are the results shown in Fig. 6 averages from all 20 selected grains? The "fraction frozen" label on the y-axis seems to indicate this. If yes: why did you select 20 individual grains if you lump all results together in the end?

The grains have not been averaged. Each droplet in the experiment contains a single grain and so there is a single freezing temperature recorded for each grain. The plot is showing the fraction frozen.

• You show 19 IR-NIPI (black squares) and 19 µL-NIPI (red circles) data points. Is there a correspondence with the individual chips? Is on each line of the plot a pair of IR-NIPI and µLNIPI results of the same chip? If this is the case, the labeling of the y-axis as fraction frozen is misleading. Rather, in this case, the y-axis should indicate a numbering of the grains.

We have include a separate plot in fig 6 to demonstrate this. We have now plotted IR-NIPI freezing temperature for a feldspar chip against the freezing temperature measured on the µl-NIPI. The one to one line is displayed along with the error in temperature for the two instruments. This can be seen in figure 6 below.

[Figure]

We have revised the text to the following in section 3.3: "The resulting fraction frozen plot for this experiment can be seen in Figure 2a and the corresponding correlation plot is shown in Figure 6b. The two instruments yielded similar fraction frozen curves and the individual feldspar grains nucleated ice at a similar temperature in both experiments. The correlation plot in Figure 6b shows that the freezing temperatures of a single grain were not identical in the two experiments, which is consistent with the stochastic nature of nucleation at active sites that have a characteristic freezing temperature (Vali, 2008; Vali, 2014). The agreement between the two instruments suggests that the temperature measurement and calibration of the IR-NIPI were robust and that there is no major temperature gradient within the aliquots in the multiwell plates."

The caption has also been updated to "**Figure 1. (A)** Plot of the fraction frozen curves for single feldspar particles per droplet in both the µL-NIPI (using 1 µL droplets) and IR-NIPI (using 50 µL droplets) experiments. The error bars display the error in temperature measurement on both instruments. **(B)** Shows the freezing temperature for the individual feldspar chips as measured by the IR-NIPI and µL-NIPI instruments. The one to one line is shown in bold and the error in temperature for the two instruments are represented by the error bars."

Why 19 and not 20 data points when you chose 20 grains?

This is a typo. We originally had 20 grains but one grain was dropped whilst transferring it over to the next experiment so only 19 grains were tested on both experiments. We have changed 20 to 19 in the text.

• What is the meaning of the error bars? Did you perform several freezing cycles and average? Did you average over several droplets on top of a grain in the case of µL-NIPI?

The Error bars are simply the errors associated with the temperature measurement. This has now been clarified in the figure caption. "**Figure 2.** Plot of the fraction frozen curves for single feldspar particles per droplet in both the µL-NIPI (using 1 µL droplets) and IR-NIPI (using 50 µL droplets) experiments. The error bars display the error in temperature measurement on both instruments."

• In some cases the error bars of IR-NIPI and µL-NIPI do not overlap. Any explanation?

This is consistent with the stochastic nature of nucleation at active sites. This is now discussed in the revised section 3.3.

• It is unclear why chips are taken and not suspensions with concentrations adjusted such that the mass of K-feldspar is the same for the 1 µl drops of the µL-NIPI and the 50 µl drops in the multiwell plate of the IR-NIPI setup.

We have improved the motivation of section 3.3. We have now added the following text "The purpose of this experiment was to have the same amount of material per droplet in each experiment and to have the material at the base of the droplet in order that the results from the two instruments could be directly compared. In doing so we could investigate the extent to which the gradient within the 50 µL wells might be a problem."

Line 259: ns(T) should be the same independent of concentration because it refers to one temperature and is per illite surface area present in a sample.

When we change the surface area of material in a droplet we change the $n_s$ to which we are sensitive. We have modified the text to refer to surface area instead of wt% suspensions to help minimise confusion. "The results demonstrated good agreement with each other and exhibited the expected trend of the droplets containing smaller amounts of nucleant surface area freezing at lower temperatures and having higher $n_s(T)$ values than the droplets with higher amounts of surface area."

Line 300: 167 L: shouldn't 16.7 x100 yield 1670 L?

This was a typo and has now been changed to 1670

Line 300: Why is "filters" in plural? How many filters did you collect and measure?

One sample was collected and referred to here. Others were collected, but these will appear in a separate publication. The text has been altered to reflect this. "In order to demonstrate the utility of this approach for atmospheric aerosol samples, a filter sample was collected in Leeds as part of a field campaign held on the evening of the 5th November. A sample of atmospheric aerosol was collected using a Mesa PQ100 air sampler for 100 min. An inlet head with an upper cut-off of 10µm was utilised and air was sampled at 16.7 L min$^{-1}$ on to a 0.4 µm polycarbonate track-etched Whatman filter, with a total of 1670 L of air sampled. The filter was then placed in to 6 mL of Milli-Q water and vortexed for 5 min to wash the particles from the filter and into suspension."

Line 310: specify what kind of modelling you mean here.

The text has been modified to "Since high-resolution regional modelling of the effect of INP on high latitude, cold sector-clouds suggests that 0.1 to 1 INP L$^{-1}$ is a critical concentration and much lower concentrations still impact clouds (Vergara-Temprado et al., 2018), measurements with IR-NIPI will be extremely useful, particularly in environments with low INP concentrations."

Lines 317 – 318: this formulation should be improved

The text has been adjusted to "We demonstrate that IR thermometry is a sound method for determining the freezing temperature of 50 µL water droplets in multiwell plates. This method overcomes potential distorting influences such as thermal gradients across the plate, the effect of freezing wells warming surrounding wells and poor thermal contact to the underlying cold plate."

Line 333: Could you specify here how you intend to automate the system further.

We have ideas for the potential automation but would rather not specify them here. We realise this comment is vague and so have removed the following sentence from the text. "The use of the multiwell plates and the IR camera lends the IR-NIPI to the possibility of automating the system further and this is an objective for future work."

Figure 1: can you indicate in this figure how the Stirling engine chiller is connected to the aluminium cold stage?

The figure has been adjusted to show this. Please see below.

[Figure]

Figure 2B: one temperature ramp seems to be an outlier to warmer temperature. Do you know why?

We think the well associated with this temperature ramp had poor thermal contact with the underlying pad. An advantage of using the IR measurement of each well is that the influence of such issues is removed. The following has been added to the caption: "Note that one well had a higher temperature than the others, likely due to poor thermal contact with the aluminium substrate. By using IR thermometry to measure the temperature of each well individually such variability is accounted for."

Figure 3, inset: the water in the aluminum well is drawn as if it did not wet the well. Is this realistic? Was the aluminum coated?

The aluminium was anodised and was hydrophobic, hence this is realistic.

Figure 3, the figure caption of panel B needs to state explicitly whether T(thermocouple) – T(IR) is shown or the opposite. You might discuss the implications of the negative peaks observed in the residuals of panel B.

This has been clarified in the figure caption "The difference was calculated by IR(T) – Thermocouple(T). The negative spikes are a result of the IR camera directly reading the water temperature as it is heated by ice formation whereas the thermocouple measurement is reading the temperature of the aluminium well which is less affected by the latent heat release."

Figure 3, figure caption, line 523: "The point of freezing is highlighted in blue"… This statement is confusing since a whole area is indicated in blue. Do you mean "the area of freezing" or to which point do you refer?

This has now been corrected to "The range over which freezing occurs is highlighted with a blue rectangle as this is where the thermal properties of ice and the initiation of heat release affect the temperature readings."

Figure 4, figure caption, line 542: "A schematic diagram of the experiment is shown of the wells within a 96 well plate chosen for temperature checks." This sentence needs to be improved.

The text has been modified to read "A diagram of the wells within a 96 well plate chosen for the comparison of IR and thermocouple measurements is displayed as an inset."

Figure 4, the figure caption of panel B needs to be improved. Which wells are shown? Can you number the wells in the multiwell plate shown as an inset in panel A according to the numbering in panel B?

This is a good idea, we have adjusted the inset to correspond to the numbers in panel B and amended the figure caption accordingly. Please see below.

[Figure]

Figure 5: the y-axis has a strange scale with 5 digits between labels, implying a spacing of 0.083333333?!?! You might want to improve.

Thank you for pointing this out. It has now been adjusted to 0.1 increments.

Figure 5: the different blank curves should be given with different colors or symbols so that they can be discriminated from each other. For some blanks, the data points at low fraction frozen seem to be missing!?

We have created 3 separate panels for this figure to show the corresponding internal blanks with the relevant experiments. The main/original plot shows a culmination of representative blanks from this set up. The data points at low fraction frozen are not missing for the blanks. For the internal water blanks 12 wells were dedicated to a handling blank, hence the lower data density compared to blanks where we used all of the wells to test the water quality.

[Figure]

Figure 7A: The two darker blues and the two darker reds are difficult to discriminate. Please improve the color palette.

This has now been amended.

Figure 7A, figure caption: the error bars should be explained. Is this the error of ±0.9°C or the standard deviation between several freezing cycles?

The figure caption has been amended to address this. "The error bars represent the temperature error of ±0.9°C."

**Technical comments**

Line 49: "systems" should be replaced by "methods"

Done

Lines 73 – 75: "This instrument is part of the NIPI suite of instruments that includes the μL-NIPI and when used together these devices allow measurements to be taken over a very wide range of INP concentrations." Consider splitting this sentence into two

Changed to "This instrument is part of the NIPI suite of instruments that includes the μL-NIPI. When used together these devices allow measurements to be taken over a very wide range of INP concentrations."

Line 92: what is meant by "other expressions"?

Changed to "The fraction of the droplet population frozen throughout the explored temperature range can then be determined, from which the ice-nucleating active site density or INP concentration can be derived (Vali et al., 2015)."

Line 130: "Stirling engine chiller" would be more precise.

This has been changed to "The unique design, in combination with a Stirling engine-based chiller…".

**Lines 156 – 158: the wording of this sentence should be improved.**

This has been adjusted to "This contrasts with the approaches adopted in other experiments where the temperature is recorded and assumed to be representative for all droplets, for example when employing a cold stage housing an embedded thermocouple whose reading is used to represent the temperature of the droplet array."

**Line 214: remove "see".**

Done

**Line 220: The reference "Polen et al., 2018" is missing from the reference list.**

Corrected

**Line 269: delete "and is illustrated"**

Done

**Line 271: "takes advantage" sounds strange. Try rewording.**

This has been reworded to "This material has also been investigated by Beall *et al.* (2017) using an instrument that also uses 50µL droplets: the Automated Ice Spectrometer (AIS)."

**Line 274: "were": you switch to past tense here, while before you used present time.**

The text has been altered to be in present. "Both the IR-NIPI and AIS data are in good agreement with one another. It can be seen that the larger volume assays (IR-NIPI and AIS) give results towards the upper spread of literature data but are still consistent with other results (**Error! Reference source not found.**b). Dry dispersed techniques have also been plotted as unfilled blue squares in Fig. 7b, but none of these techniques are sensitive in the range of $n_s(T)$ seen by the large droplet instruments. The new data from the IR-NIPI has extended the dataset for NX-illite to warmer temperatures than in previous measurements, illustrating the utility of the technique."

**Line 281: "flowing": do you mean "following"?**

Yes, thank you. This has been changed.

**Line 288: "suspended" instead of "suspend".**

Done

**Line 300: "in to" should be one word.**

Done

**Line 305: in the equation it is "Nu(T)" not "Nυ(T)". Please adjust.**

Done

**Lines 328 – 329: "an aerosol sample in the atmosphere of the city of Leeds" sounds strange. Improve formulation to e.g.: "an aerosol sample from the City of Leeds".**

This has been amended to "The utility of IR-NIPI for the analysis of atmospheric samples was also demonstrated by collecting and analysing an aerosol sample from the city of Leeds, England."

**Line 453 – 454: use abbreviation: Atmos. Chem. Phys.**

Done

**Legend of Fig. 6: "IR-NIPI" not "IR-NIP"**

Thank you. This is now fixed

---

## Author Comment (AC3) · 6 Aug 2018

**Response to Referee #3**

We would like to thank the referee for their insightful comments and have responded below. The referee comments are highlighted in red with our responses in black.

This is a well-written paper with an interesting experimental approach that fits perfectly well into the journal Atmospheric Measurement Techniques. The authors describe an instrument for quantifying heterogeneous ice nucleation: the InfraRed-Nucleation by Immersed Particles Instrument (IR-NIPI). They use multiwell plates and an infrared camera for detection of the freezing process. For comparison, they have investigated homogeneous ice nucleation of ultrapure water, and the heterogeneous freezing of two mineral dust samples. The manuscript should be published in AMT after major revisions.

**Comments**

In line 72, the authors claim "Here we propose a new technique,. . .". Unfortunately, this is not entirely true. A quick literature research shows that there are other instruments with very similar approaches. In particular, I would like to mention the set-ups of Zaragotas et al. and of Kunert et al. It is good scientific practice to search, to describe, and to discuss the findings of other scientists when presenting a new set-up. I expect that the authors make up the leeway in the revised version of the manuscript. Concerning the set-up of Kunert et al., I could not find any peer-reviewed publication, but I have been the organizer of two ice workshops (Kunert 2016b, 2017b) and the convener of two EGU General Assembly sessions (Kunert 2016a, 2017a) and a speaker at the INUIT Final Conference and 2nd Atmospheric Ice Nucleation Conference (Kunert 2018), where this research has been presented. At all these meetings, also the authors were present and in the case of the latter have even been the organizers. Therefore, the Twin-plate ice nucleation assay (TINA) with infrared detection by Kunert et al. is well-known to them and should be described in their manuscript for comparison with IR-NIPI.

The omission of Zaragotas *et al*. (2016) was a major oversight and we thank the reviewer for bringing this to our attention. The instrument discussed is applied to the cryobiology field but is a similar set up to the IR-NIPI and supports the use of IR cameras for this application.

With regards to the various conference presentations made by Kunert et al. there was no peer reviewed literature for this technique at the time of submission and hence it was not cited or discussed. However on the 24th of July Kunert et al. have had a publication accepted for review and posted on the AMT discussion forum. We are then happy to include this reference and have altered the text to include it.

We have included the references to these papers in sections 1 and 2.4.

In section 1: "While many instruments use optical cameras to detect freezing events (Whale et al., 2015; Budke and Koop, 2015; Häusler et al., 2018; Beall et al., 2017), some researchers have used techniques to detect the release of latent heat associated with freezing. For example differential scanning calorimetry (Marcolli et al. 2007; Pinti et al. 2012) and infrared emissions (Zaragotas et al., 2016; Kunert et al. 2018) have been used. Zaragotas *et al*. (2016) use a thermal camera to measure the temperature of individual aliquots within a 96 multiwell plate partially submerged within an alcohol bath. This study investigated plant samples but suggested that the technique may be adapted for atmospheric purposes. Very recently, Kunert *et al*. (2018) presented a similar set up to investigate biological samples and collected aerosol. Unlike Zaragotas *et al*. (2016), Kunert *et al.* (2018) do not measure individual droplet temperatures via infrared emissions but instead use multiple thermistors embedded in the sample holders to infer temperature for the droplet array."

In section 2.4: "It should be noted that one of the limitations of the setup used by Zaragotas *et al*. (2016) was that the IR camera was calibrated only once by the factory, however our calibration method mitigates this limitation."

In line 48, the authors list some droplet freezing assay experiments but the list is rather incomplete, e.g. Häusler 2018 is missing. I strongly recommend a table with all technical parameters of each experiment listed, e.g. number of observed volumes, volume of the droplets, homogeneous ice nucleation temperature, etc. Finally, for all experiments a discussion of the pros and cons in comparison to IR-NIPI should be added.

This list was not intended to be exhaustive, but representative. We have now added the reference of Häusler *et al*. (2018) to the list of drop assay references.

We think that a table reviewing all previous droplet freezing assays is beyond the scope of this techniques paper. A table of such a nature would be much better suited in a review or intercomparison paper. There are papers already available which describe such advantages and disadvantages of the various styles of instruments and are referenced in the text "For more information on the capabilities and limitations of the various techniques see the comprehensive reviews and intercomparisons conducted by Hiranuma *et al*. (2015) and (DeMott et al., 2018). Haüsler *et al*. (2018) also presents a summary of the features of various techniques. ". In addition, we are aware of a paper describing many techniques from the FIN02 activities which has recently been submitted to AMTD and added to the references in text.

Instead, the authors compare only their own set-ups, i.e. µL-NIPI and IR-NIPI. However, the volume of the respective droplets is very different, 1µL versus 50µL, respectively.

We have compared a wide variety of different techniques when investigating the NX-illite sample. We chose NX-illite as it allowed a direct comparison to literature data of a similar experiment in terms of droplet volume (the AIS instrument) as well as a wide range of other instruments (Hiranuma *et al*., 2015). This included a direct comparison to another technique using 50 µL droplets (AIS). It should be noted there is limited literature data for 50µL droplets and NX-illite is the only material we can compare directly to as a result of this. We compared to our ul-NIPI technique for other samples, because this is available in our laboratory and we could operate the two instruments side by side with the same sample.

This is not only important for homogeneous ice nucleation, which shows strong volume dependence, but also is important for heterogeneous ice nucleation because larger volumes carry more INPs and the abundance of efficient INPs rises. The authors have discussed this only partly and a more elaborated discussion might be necessary.

The text has now been amended to expand on this topic to read "The alternative approach is therefore to increase the number of particles within each aliquot of water. In principle, increasing the number of particles per droplet, and therefore the surface area of nucleator per droplet, will increase the sensitivity of the experiment to rarer INP. This enables quantification of lower INP concentrations" To increase the number of aerosol particles per volume of liquid the time period over which an atmospheric sample is collected can be extended, but in doing so temporal resolution would be lost. A method of increasing the sensitivity of an immersion mode technique is to increase the volume of the collected suspension used in each aliquot, while maintaining the concentration of particles per unit volume. This increases the number of particles per aliquot of liquid and therefore makes it more likely that rarer INP will be detected."

In particular, I miss plots of the homogeneous freezing events and a detailed study of the freezing of single droplets (marked with numbers on a picture of the multiwell assay). I also recommend adding the diameter of the droplets to the volume to make the study more comparable to other studies.

We are unable to reach homogenous in this setup, as with all techniques employing such large droplets. The expected homogeneous curve can be seen in figure 5 and our blanks are well above this. We do not see the value of numbering specific droplets in an array, this is not information we routinely collected.

Our droplets are not spheres in the multiwell plates so a diameter is not relevant but we have quoted a volume equivalent diameter to help compare to other studies. "The most useful for this freezing assay are the 96 x 200 µL or 384 x 50 µL aliquot arrays and in the tests reported here 50µL droplets (~2300µm volume equivalent diameter) are used in 96 well plates."

The authors make the point that their set-up is more sensitive for low concentrations of INPs, which is particularly true for strong INPs. However, they don't mention the disadvantage of their set-up, which is that they cannot easily measure weak INPs. In the atmosphere, the number of strong INPs is extremely low, which makes µL-NIPI a valuable technique. However, often strong INPs are entirely missing and weak INPs will be much more abundant. Therefore, the authors should discuss the limitations of their set-up and should also show experiments at the detection threshold and should investigate proxies for weak INPs e.g. cellulose or soot.

We have now tried to clarify the point that the IR-NIPI instruments compliment smaller droplet techniques. The section of text added reads "These large volume assays capture the rarer, more active INP but often miss the

more abundant but less active INP. Hence they should ideally be used alongside a smaller droplet instrument to generate complimentary datasets."

Also I miss biological INPs or proteins and polysaccharides been emitted by biological sources. Therefore, beside ns values of solid INPs also nm values of soluble INPs should be measured and discussed.

We accept that this would be interesting to investigate but again believe it is beyond the scope of this paper which is to outline the concept of the technique. We have some interesting results from biological materials and would like to use these in another paper and do not think this is the best place to present this data. We believe that we have already provided sufficient examples of materials used in the IR-NIPI system.

**Specific comments**

Where is the homogeneous freezing temperature (T50) of ultrapure water in your IRNIPI set-up?

Homogenous freezing (T50) for 50µL droplets is ~-32.5°C based on Murray and Koop (2016). This is presented in figure 5. We have not accessed homogeneous freezing and do not claim to do so.

Line 99: Also indicate the formula for nm and add respective water soluble samples.

We do not think this is necessary as explained above.

Line 182: "standard deviation ±0.5∘C"

Thank you. This has been changed.

Line 190: "after the first equilibrium step at +5∘C"

Thank you. This is changed to the suggested.

How is the temperature uncertainty in the range between -20∘ and -30∘ C?

We are unable to quantify the uncertainty as the thermocouples and water often freezes before this point but we assume that it is similar to that stated for the temperature range above -20°C. We are mainly interested in the temperature of samples above -20°C as below this we enter our baseline. The uncertainty within this temperature range is of secondary importance for our experiments.

You have only used ultrapure water for temperature calibration. How about other samples such as aqueous salt solutions, higher alcohols or alkanes?

As you have rightly pointed out we have only calibrated on the basis of ultrapure water. We have done this because we are only using water suspensions and have not attempted to study nucleation in different solutions or other materials. Furthermore, using solutions or other liquids for a calibration would not be applicable for water, since they have different thermal conductivities and different thermal emissivity. The IR camera is already corrected based on the thermal emissivity of water and hence using other solutions or liquids would introduce new errors.

Line 215: What kind of filter has been used for purification?

Thank you for highlighting this. It has now been added to the text. "Sartorius Ministart, non-pyrogenic, single use filters were used for this (product code 17597-K)."

A figure, similar to that in fig. 7B, should be plotted also for feldspar samples including comparison data from other groups.

We believe that is not needed for this paper as for the same reasons as mentioned earlier. We believe we have already demonstrated the use of the IR-NIPI with sufficient materials and that this would be more applicable for an intercomparison paper and has been done so by DeMott *et al.* (2018). In addition there is also no published literature data for feldspar in the droplet regime for the IR-NIPI so no direct comparison can be made. This was one of the reasons we chose to include a comparison with NX-illite.

In figures 2, 3, 4, and 7 capital letters have been used in the graph but small letters have been used in the figure caption, respectively.

Thank you. This has been amended.